# A Hierarchical Adaptive Moment Matching Multiple Model Tracking Method for Hypersonic Glide Target Under Measurement Uncertainty

**DOI:** 10.3390/s25216621

**Published:** 2025-10-28

**Authors:** Hanxing Shao, Jibin Zheng, Yanwen Bai, Hongwei Liu, Ye Ge, Boyang Liu

**Affiliations:** 1National Key Laboratory of Radar Signal Processing, Xidian University, Xi’an 710071, China; 23021211470@stu.xidian.edu.cn (H.S.); yanwenbai@stu.xidian.edu.cn (Y.B.); hwliu@xidian.edu.cn (H.L.); 2Reentry Dynamics and Target Characteristic Laboratory, Unit 63610, People’s Liberation Army of China, Korla 841001, China; geyedli@163.com (Y.G.); liuby5010743@sina.com (B.L.)

**Keywords:** hypersonic glide target, radar target tracking, quasi-equilibrium glide, variable structure multiple model, robust minimum error entropy criterion

## Abstract

Hypersonic glide targets (HGTs) pose significant challenges for radar tracking due to complex maneuver strategies and time-varying statistics of measurement noise. Conventional single-model tracking methods are generally insufficient to fully capture maneuver modes, while existing multiple-model methods face trade-offs between model set completeness and computational efficiency. In addition, existing tracking methods struggle to cope with the non-Gaussian noise during hypersonic flight. To overcome these limitations, a Hierarchical Adaptive Moment Matching (HAMM) multiple-model method is proposed in this paper. Firstly, a comprehensive model set is constructed to cover characteristic maneuver modes. Subsequently, a hierarchical multiple-model framework is developed where: (1) a coarse model set is dynamically adapted by multi-frame posterior probability evolution and Rényi divergence criteria; (2) a fine model set is generated based on the moment matching method. Furthermore, the minimum error entropy cubature Kalman filter (MEECKF) is proposed to suppress the non-Gaussian measurement noise with high stability. Monte Carlo simulations demonstrate that the proposed method achieves improved positioning accuracy and faster convergence.

## 1. Introduction

HGTs pose significant challenges to national defense due to their high speed and complex maneuver modes [1,2]. During HGT’s reentry mission, the aerodynamic configuration with high lift-to-drag ratios enables them to execute a wide range of maneuvering strategies [3,4]. However, the maneuvering start time, acceleration, and maneuvering frequency are usually unknown. Multiple-model (MM) state estimation provides an effective framework for addressing this challenge [5].

The selection of tracking models is critical in MM methods. Tracking models could be generally categorized into two types: dynamic models and kinematic models [6]. Dynamic models are based on the target’s aerodynamic parameters and force characteristics. The design of dynamic models typically requires substantial prior information and yields unsatisfactory results when aerodynamic parameters are time-varying [7]. The kinematic models describe the maneuver of targets through a reasonable stochastic process assumption. Commonly used kinematic models include constant velocity, constant acceleration, current statistical, and jerk models. Recently, various specialized models have been proposed to characterize the motion characteristics of HGTs during the glide phase: Wang et al. [8] proposed the sine wave model to characterize periodic oscillations in the trajectory. Li et al. [9] proposed the zero-mean damped oscillation model to describe the damped properties of the acceleration autocorrelation function, thereby providing a damped oscillation model. Due to continuous aerodynamic forces in hypersonic glide, HGTs exhibit non-zero mean acceleration, and Cheng et al. [10] further proposed the non-zero mean damped oscillation model. This model separately characterizes disturbance and mean terms for accurate acceleration estimation. However, it lacks a definitive parameter configuration scheme, which may degrade tracking performance under model mismatch.

In practical applications, MM methods provide a systematic, convenient and powerful solution for hybrid system estimation. MM methods are primarily categorized into three classes: autonomous MM, cooperating MM (CMM), variable-structure MM (VSMM) [11]. In the CMM class, the interactive multiple model (IMM) algorithm has seen extensive applications [5,12,13,14,15,16]. However, the IMM algorithm maintains a fixed model set throughout filtering, impairing its adaptability to targets with frequently switching motion modes. To cover all potential motion modes, IMM typically expands its model set. However, excessive expansion of the model set induces model competition, which will instead reduce the tracking accuracy and substantially increase computational load. To overcome this limitation, Li et al. introduced the third-generation VSMM approach [17]. VSMM dynamically adjusts both model set structures and parameters, with three representative strategies: model group switching (MGS) [18], likely model set (LMS) [19], and expected mode augmentation (EMA) [20]. Although these strategies perform robustly in conventional target tracking, they face the following challenges when applied to HGTs: (1) MGS dynamically activates/deactivates predefined model sets, but its effectiveness depends on the initial model set partitioning and topological configuration. Given the high-dimensional model parameter space of HGTs tracking models, designing a practical topological structure is challenging. (2) LMS enhances adaptability through dynamic model set adjustments, yet its per-time-step model updates degrade convergence. (3) EMA adapts to parametric variations through generative expansion of tunable models, but cannot accommodate structural model changes. As an enhanced variant of LMS, the best model augmentation (BMA) method [21] employs Kullback–Leibler (KL) divergence for refined model selection but inherits LMS’s static model set limitation, where base and candidate models lack parametric adaptation to maneuver characteristics, and it introduces prohibitive computational overhead due to per-step KL divergence recomputation. Liu et al. [22] developed the hybrid grid IMM (HGIMM) algorithm to augment the original model set. HGIMM constructs a fine-grid model set at time *k* by recombining parameters from a fixed coarse-grid set at k−1 for localized parameter adaptation. However, the fixed coarse model set remains inadequate to cover maneuver modes of HGTs, particularly under phase-transition scenarios.

Besides the tracking model, it is also crucial to select a suitable filtering algorithm to suppress the measurement noise. The initial derivation of the Kalman filter (KF) stemmed from a linear state-space model assuming a Gaussian distribution [23,24]. In practice, the relationship between measurement and state is nonlinear. Several extensions of KF are derived to address nonlinear problems, including extended KF, unscented KF, cubature KF (CKF), and others [25,26,27]. CKF is a preferred method due to its superior precision compared to the extended KF and unscented KF, while maintaining moderate computational costs.

Most of these KFs are derived by the minimum mean square error (MMSE) criterion [28]. They may encounter filter performance degradation when dealing with complex noises due to the inadequacy of the MMSE criterion under non-Gaussian noise [29,30,31,32,33]. A particle filter (PF) can handle non-Gaussian measurement noise by approximating the posterior distribution through weighted particles [34]. However, PF suffers from severe particle degeneracy in high-dimensional systems, requiring an exponentially growing number of particles [35]. While Rao–Blackwellized PF reduces this requirement, it demands partial linear measurement relationships, which are not satisfied in radar tracking with range and angle measurements [36,37]. Therefore, we focus on Kalman-type filters that better suit the radar setting.

Recently, information theory has been utilized for state estimation under non-Gaussian measurement noise and has yielded promising results. The minimum error entropy (MEE) criterion exhibits superior robustness due to its strong capability in modeling error entropy [38,39]. Therefore, MEE-based KF and MEE-based extended KF have been developed to address more complex noise, such as multi-modal distribution noise. However, the MEE-based extended KF has restricted tracking accuracy since its derivation involves the linearization of a nonlinear function. Ref. [40] proposed a MEE filter algorithm based on the CKF framework to solve this problem. But this method may exhibit numerical instability due to the potential singularity of the weight matrix when confronted with significant outliers [41].

The preceding discussion reveals three main challenges: a lack of systematic design approaches for HGT’s kinematic model set; limitations in existing VSMM methods for HGTs tracking; and insufficient research on robust filters within VSMM frameworks. To overcome these challenges, this paper introduces the HAMM multiple model method and MEECKF. The key contributions and improvements over existing results can be summarized as follows.

1.Unlike existing works that focus on single specialized models (e.g., sine wave model [8], damped oscillation models [9,10]), we propose a comprehensive design methodology that systematically integrates multiple kinematic models to cover the HGTs’ motion modes. This ensures better coverage of diverse maneuver modes.2.Compared to existing VSMM approaches—MGS’s dependency on initial topology design, LMS’s convergence issues due to per-step updates [19], EMA’s inability to handle structural changes [20], and BMA’s prohibitive computational cost [21]—our HAMM algorithm introduces an adaptive strategy. It performs time-varying updates to the primary model set while dynamically generating optimized candidate sets through moment matching. This dual-layer design significantly improves maneuver response speed and filtering consistency while maintaining computational efficiency, particularly for HGTs with frequently switching motion modes.3.While MEECKF has demonstrated robustness under non-Gaussian measurement noise [40], existing VSMM-based HGTs tracking methods rely on standard Kalman-type filters, which suffer performance degradation under complex noise conditions. Our work is to integrate MEECKF into a VSMM framework for HGTs tracking, combining the robustness of the MEE criterion against non-Gaussian noise with the adaptability of variable-structure multiple models. This integration addresses the critical gap in handling uncertain measurement noise statistics within adaptive multiple-model tracking systems.

The remainder of the article is organized as follows. Section 2 introduces the tracking model and the construction of the model set. Section 3 describes the design of the HAMM algorithm. Section 4 designs the robust filter. Section 5 presents the simulation results. Section 6 summarizes the conclusions of this study.

## 2. Tracking Model Set

### 2.1. Tracking Model

During the gliding phase, HGT is predominantly influenced by the gravitational force of Earth and aerodynamic forces. Ignoring the effect of the earth’s rotation [42], the nonlinear dynamical equation of HGT is represented by(1)$$\left\{\begin{array}{c} \frac{dr}{dt}=v\sin\gamma \\ \frac{d\theta }{dt}=\frac{v\cos\gamma \sin\psi }{r\cos\phi }\\ \frac{d\phi }{dt}=\frac{v\cos\gamma \cos\psi }{r}\\ \frac{dv}{dt}=-\frac{D}{m}-g\sin\gamma \\ \frac{d\gamma }{dt}=\frac{1}{v}\left[\frac{L\cos\sigma }{m}+\left(\frac{v^{2}}{r}-g\right)\cos\gamma \right]\\ \frac{d\psi }{dt}=\frac{1}{v}\left[\frac{L\sin\sigma }{m\cos\gamma }+\frac{v^{2}}{r}\cos\gamma \sin\psi \tan\phi \right] \end{array}\right.$$
where *r* is the position vector from the center of the earth to HGT. θ and ϕ represent the longitude and latitude, respectively. *v* stands for the velocity. γ and ψ denote the flight path angle and heading angle, respectively. *D* and *L* are the drag force and lift force, respectively. *m* and *g* denote the aircraft mass and gravitational acceleration, respectively. σ represents the bank angle of HGT.

While (Equation 1) describes the aerodynamic dynamics of HGT governed by lift and drag forces, the subsequent kinematic model employed for state estimation is derived from the geometric relationships of the velocity vector in the Earth-centered coordinate system. The consistency between these two formulations lies in their treatment of the velocity components and angular rates. The kinematic formulation is preferred as it avoids the need for precise aerodynamic coefficients and bank angle control inputs, which are difficult for ground-based tracking systems. Instead, the temporal evolution of the state is modeled stochastically using first-order Markov processes with oscillatory autocorrelation, which effectively captures the maneuvering characteristics of HGT while maintaining computational tractability for real-time filtering applications.

The state vector is constructed by(2)$$x_{k}=\left[\theta_{k},\phi_{k},h_{k},v_{k},a_{k},j_{k},\gamma_{k},\omega_{\gamma ,k},\psi_{k},\omega_{\psi ,k}\right]$$
where *a* is the target velocity and acceleration; *j* denotes the jerk (time derivative of acceleration); γ and $\omega_{\gamma }$ denote the flight-path angle and its angular rate; and ψ and $\omega_{\psi }$ represent the heading angle and its angular rate of the target. The change in longitude, latitude, and altitude caused by velocity is shown in Figure 1.

The variations in latitude ϕ, longitude θ, and altitude *h* are primarily determined by the velocity *v*, flight-path angle γ, and heading angle ψ, and their relationship can be expressed as(3)$$\dot{\theta }=\frac{v_{E}}{\left(R_{e}+h\right)\cos\phi }$$(4)$$\dot{\phi }=\frac{v_{N}}{R_{e}+h}$$(5)$$\dot{h}=v_{U}$$(6)$$v_{E}=v\cos\gamma \sin\psi$$(7)$$v_{N}=v\cos\gamma \cos\psi$$(8)$$v_{U}=v\sin\gamma$$
where $v_{E}$ denote the eastward velocity, and $v_{N},v_{U}$ are the northward velocity and the upward velocity, respectively. $R_{e}$ is the earth radius.

The acceleration *a*, the flight-path angle γ, and the heading angle ψ are each modeled as first-order Markov processes with exponentially decaying oscillatory autocorrelation functions of the form(9)$$R_{i}\left(\tau \right)=\sigma_{i}^{2}e^{-\alpha_{i}\left|\tau \right|}\cos\left(\beta_{i}\tau \right),\;i\in \left\{a,\gamma ,\psi \right\},$$
where $\alpha_{i}$ and $\beta_{i}$ denote the damping factor and the oscillation angular frequency, respectively, and $\sigma_{i}^{2}$ is the variance of the corresponding state variable.

By applying the Wiener–Kolmogorov whitening procedure, the state dynamics can be reformulated in terms of white Gaussian noise. For a state variable x∈{a,γ,ψ} with its derivative $\dot{x}$, the unified state equation is(10)$$\left\{\begin{array}{cc} \dot{x} & =\dot{x},\\ \ddot{x} & =-\left(\alpha_{x}^{2}+\beta_{x}^{2}\right)x-2\alpha_{x}\dot{x}+\left(\sqrt{\alpha_{x}^{2}+\beta_{x}^{2}}-2\alpha_{x}\right)w_{x}, \end{array}\right.$$
where $w_{x}$ is Gaussian white noise. Specifically, for the acceleration *a*, we have $\dot{x}=j$ (jerk) with parameters $\alpha_{a}$ and $\beta_{a}$; for the flight-path angle γ, we have $\dot{x}=\omega_{\gamma }$ (angular rate) with parameters $\alpha_{\gamma }$ and $\beta_{\gamma }$.

The heading angle ψ follows the same formulation, with corresponding parameters $\left(\alpha_{\psi },\beta_{\psi },\sigma_{\psi }^{2}\right)$ and process noise $w_{\psi }$.

The state equation is expressed as(11)$${\dot{x}}_{k}=f\left(x_{k-1}\right)+Gw_{k-1}$$
where $w_{k-1}={\left[w_{a},w_{\gamma },w_{\psi }\right]}^{T}$, f(·) is given by(12)$$\left\{\begin{array}{c} {\dot{\theta }}_{k}=\frac{v_{k-1}\cos\gamma_{k-1}\sin\psi_{k-1}}{\left(R_{e}+h_{k-1}\right)\cos\phi_{k-1}}\\ {\dot{\phi }}_{k}=\frac{v_{k-1}\cos\gamma_{k-1}\cos\psi_{k-1}}{\left(R_{e}+h_{k-1}\right)}\\ {\dot{h}}_{k}=v_{k-1}\sin\gamma_{k-1}\\ {\dot{v}}_{k}=a_{k-1}\\ {\dot{a}}_{k}=j_{k-1}\\ {\dot{j}}_{k}=-\left(\alpha_{a}^{2}+\beta_{a}^{2}\right)a_{k-1}-2\alpha_{a}j_{k-1}\\ {\dot{\gamma }}_{k}=\omega_{\gamma ,k-1}\\ {\dot{\omega }}_{\gamma ,k}=-\left(\alpha_{\gamma }^{2}+\beta_{\gamma }^{2}\right)\omega_{\gamma ,k-1}-2\alpha_{\gamma }\omega_{\gamma ,k-1}\\ {\dot{\psi }}_{k}=\omega_{\psi ,k-1}\\ {\dot{\omega }}_{\psi ,k}=-\left(\alpha_{\psi }^{2}+\beta_{\psi }^{2}\right)\omega_{\psi ,k-1}-2\alpha_{\psi }\omega_{\psi ,k-1} \end{array}\right.$$ Taking longitude θ as an example, its state prediction equation can be expressed as(13)$$\theta_{k|k-1}=\theta_{k-1}+\dot{\theta_{k}}T$$

The autocorrelation of the noises is $E\left[w_{k}w_{k}^{T}\right]=\mathrm{diag}\left[\sigma_{a}^{2},\sigma_{\gamma }^{2},\sigma_{\psi }^{2}\right]$ and the process noise covariance matrix Q is obtained as $GE\left[w_{k}w_{k}^{T}\right]G^{T}$, where G is given by(14)$$G=\left[\begin{array}{c} \begin{array}{ccc} & 0_{5\times 3} & \end{array}\\ \begin{array}{ccc} 1 & 0 & 0 \end{array}\\ \begin{array}{ccc} \sqrt{\alpha_{a}^{2}+\beta_{a}^{2}}-2\alpha_{a} & 0 & 0 \end{array}\\ \begin{array}{ccc} 0 & 1 & 0 \end{array}\\ \begin{array}{ccc} 0 & \sqrt{\alpha_{\gamma }^{2}+\beta_{\gamma }^{2}}-2\alpha_{\gamma } & 0 \end{array}\\ \begin{array}{ccc} 0 & 0 & 1 \end{array}\\ \begin{array}{ccc} 0 & 0 & \sqrt{\alpha_{\psi }^{2}+\beta_{\psi }^{2}}-2\alpha_{\psi } \end{array} \end{array}\right].$$
and $0_{5\times 3}$ denotes a 5×3 matrix with all elements equal to zero.

Radar measurements include the slant range $\rho_{r}$, the azimuth angle $\varphi_{a}$, and the elevation angle $\theta_{e}$. The measurement vector is constructed by(15)$$z_{k}={\left[\varphi_{a},\theta_{e},\rho_{r}\right]}^{T}.$$

The measurement model is expressed as(16)$$z_{k}=h\left(x_{k}\right)+v_{k}$$(17)$$h\left(\cdot \right)=\left[\begin{array}{c} \arctan\frac{y_{n}}{x_{e}}\\ \arctan\frac{z_{u}}{\sqrt{x_{e}^{2}+y_{n}^{2}}}\\ \sqrt{x_{e}^{2}+y_{n}^{2}+z_{u}^{2}} \end{array}\right.$$
where $\left[x_{e},y_{n},z_{u}\right]$ represents the coordinate components of the target in the radar’s East-North-Up(ENU) coordinate system.

Let the radar be located at $\left[\theta_{r},\phi_{r},h_{r}\right]$. Then, the relationship between the target’s longitude θ, latitude ϕ, altitude *h* and its coordinates in the radar’s ENU frame $\left(x_{e},y_{n},z_{u}\right)$ can be expressed as(18)$$\left[\begin{array}{c} x_{e}\\ y_{n}\\ z_{u} \end{array}\right]=\left[\begin{array}{c} -\left(R_{E}+h\right)\cos\phi \sin\left(\theta -\theta_{r}\right)\\ \left(R_{E}+h\right)\cos\phi_{r}\sin\phi -\left(R_{E}+h\right)\cos\phi \cos\left(\theta -\theta_{r}\right)\\ \left(R_{E}+h\right)\sin\phi -\left(R_{E}+h_{r}\right)\sin\phi_{r} \end{array}\right],$$
where $R_{E}$ denotes the Earth’s radius.

### 2.2. Construction of Model Set

A single fixed-parameter tracking model is insufficient to cover the possible motion patterns of the HGTs. To characterize the multiple maneuver modes exhibited by HGTs, this paper develops a maneuver model set based on the proposed model. The model set construction adheres to the following assumptions:1.The target’s acceleration components in each spatial dimension are independently modeled by the proposed model. The maneuver modes are categorized into three distinct dynamic modes in each dimension:N-state (no maneuver): Acceleration dominated by low-amplitude stochastic perturbations;R-state (random maneuver): Non-periodic accelerations induced by penetration maneuver;P-state (periodic maneuver): Sustained oscillatory motion typified by jump glide behavior.Representative trajectories corresponding to the N, R, and P states are illustrated in Figure 2.2.Parametric controllability is achieved through $\alpha_{i},\;\beta_{i},\;\sigma_{i},\;i\in \left\{a,\gamma ,\psi \right\}.$Maneuver frequency $\alpha_{i,l}$: Characterizes slow-varying disturbances during gliding; $\alpha_{i,s}$: Sustained high-intensity maneuvering.The angular rate of the target periodic maneuver $\beta_{i}$:$\beta_{i,s}\to 0$: Non-periodic equivalence. $\beta_{i,l}$: Signature jump glide frequency.State variance $\sigma_{i}$: The magnitude of acceleration variance provides a direct measure of target maneuver intensity: larger variance $\sigma_{i,l}$ typically indicates more rapid and intense maneuvers, while smaller variance $\sigma_{i,s}$ reflects smoother motion.

The one-dimensional tracking model can be summarized in Table 1.

Diverse maneuver modes can be modeled through three-dimensional state combinations. A tri-letter coding method denotes the state configuration per spatial dimension, as illustrated below:Non maneuver: NNN.Uni-directional random maneuver: RNN, NRN, and NNR.Uni-directional periodic maneuver: PNN, NPN, and NNP.Bi-directional random maneuver: RRN, RNR, and NRR.Bi-directional periodic maneuver: PPN, PNP, and NPP.Bi-directional hybrid maneuver: PRN, PNR, NPR, RPN, RNP, and NRP.Tri-directional hybrid maneuver: RRR, PRP, PPR, RPP, PRR, and RPR.

Although certain state combinations are theoretically enumerable, their physical realizability remains constrained. For instance, the triple periodic maneuver state (PPP) exhibits limited physical significance in practical scenarios. Other combinations are physically reasonable, but they overlap with the corresponding models. To reduce the computational complexity, such combinations are excluded.

## 3. HAMM Multiple-Model Method

### 3.1. Framework of HAMM

Define the mode space S comprising all possible system modes, where for true motion mode $s_{k}$ satisfies $s_{k}\in S$. The state Equation (Equation 11) and measurement Equation (Equation 16) are characterized by a hybrid system:(19)$$x_{k}=F_{k}\left(s_{k}\right)x_{k-1}+G_{k}\left(s_{k}\right)w_{k-1}\left(s_{k}\right)$$(20)$$z_{k}=h\left(s_{k},x_{k}\right)+v_{k}\left(s_{k}\right)$$

In operational scenarios, $s_{k}$ cannot be precisely determined and is typically approximated by models. By constructing a model set $M={\left\{m^{\left(j\right)}\right\}}_{j=1}^{N}$, (Equation 19) and (Equation 20) are approximated by:(21)$$x_{k}=F_{k}\left(m_{k}^{\left(j\right)}\right)x_{k-1}+G_{k}\left(m_{k}^{\left(j\right)}\right)w_{k-1}\left(m_{k}^{\left(j\right)}\right)$$(22)$$z_{k}=h\left(m_{k}^{\left(j\right)},x_{k}\right)+v_{k}\left(m_{k}^{\left(j\right)}\right)$$

In the HAMM algorithm, M comprises the coarse model set $M^{c}$ and the fine model set *A*. For $M_{k}$ at time *k*, it holds that(23)$$M_{k}=M_{k}^{c}\cup A_{k}$$

The state estimation can be expressed as:(24)$$\begin{array}{cc} {\hat{x}}_{k} & =E\left[x_{k}|M^{k},z_{k}\right]\\ \; & =E\left[x_{k}|M_{k}^{c},M^{k-1},z_{k}\right]\mathrm{Pr}\left(M_{k}^{c}|M^{k},z_{k}\right)\\ \; & \;+E\left[x_{k}|A_{k},M^{k},z_{k}\right]\mathrm{Pr}\left(A_{k}|M^{k},z_{k}\right) \end{array}$$

Define the following variables:(25)$$\begin{array}{cc} \; & \mu_{k}^{M_{k}^{c}}=\mathrm{Pr}\left\{M_{k}^{c}|M^{k},z_{k}\right\}\;\mu_{k}^{A}=\mathrm{Pr}\left\{A_{k}|M^{k},z_{k}\right\}\\ \; & {\hat{x}}_{k}^{M_{k}^{c}}=E\left[x_{k}|M_{k}^{c},M^{k-1},z_{k}\right]{\hat{x}}_{k}^{A_{k}}=E\left[x_{k}|A_{k},M^{k-1},z_{k}\right] \end{array}$$
where $\mu_{k}^{M_{k}^{c}}$ and $\mu_{k}^{A_{k}}$ denote the global posterior probabilities at time *k* for the coarse model set $M_{k}^{c}$ and fine model set $A_{k}$, respectively; ${\hat{x}}_{k}^{M_{k}^{c}}$ and ${\hat{x}}_{k}^{A_{k}}$ represent the state estimates obtained at time *k* based on $M_{k}^{c}$ and $A_{k}$, respectively. Substitute (Equation 25) into (Equation 24) yields:(26)$${\hat{x}}_{k}=\mu_{k}^{M_{k}^{c}}{\hat{x}}_{k}^{M_{k}^{c}}+\mu_{k}^{A_{k}}{\hat{x}}_{k}^{A_{k}}$$

Therefore, the problem is transformed into the design of $M_{k}^{c}$ and $A_{k}$.

### 3.2. Adaption of Coarse Model Set

The HAMM algorithm dynamically terminates persistently underperforming coarse models by monitoring the temporal evolution of their posterior probabilities, while optimally activating new candidates from Section 2.2 based on Rényi information divergence. Prior to our discussion, the following model sets are defined:$M_{t}^{c}$: total coarse model set (corresponding to Section 2.2).$M_{k}^{c}$: coarse model set used for state estimation at time *k*.$M_{k}^{fa}$: candidate model set for activation at time *k*.$M_{k}^{a}$: activated model set at time *k*.$M_{k}^{t}$: terminated model set at time *k*.$M_{k}^{r}$: reserved model set at time *k*.
where $M_{t}^{c}=M_{k}^{c}\cup M_{k}^{fa}$, $M_{k}^{a}\subseteq M_{k}^{fa}$, $M_{k-1}^{c}=M_{k}^{t}\cup M_{k}^{r}$, and $M_{k}^{r}\cap M_{k}^{a}=⌀$.

Performing model termination and activation at every time step may induce instability. During the initial filtering, coarse models exhibit significant short-term fluctuations in posterior probabilities. Relying solely on single-frame probability values to determine model validity at this stage risks premature termination. Moreover, frequent model activation substantially increases computational load and impedes filter convergence. Consequently, the termination and activation should be evaluated based on posterior probabilities over multiple time steps.

$M_{k}^{t}$ is defined as follows: For a model $m^{i}\in M_{k-1}^{c}$ that was activated at time step $k_{0}$, consider its posterior probability sequence from $k_{0}$ to the current time *k* denoted as $\left\{\mu_{i}^{k_{0}},\mu_{i}^{k_{0}+1},\cdots ,\mu_{i}^{k}\right\}$. If there exists a sequence of $\tau_{f}$ consecutive frames starting from $k_{0}$ such that(27)$$\mu_{i}^{\left(k^{\prime }\right)}<\mu_{0},\forall k^{\prime }\in \left\{k_{0},k_{0}+1,\ldots ,k_{0}+\tau_{f}-1\right\}$$
then $m^{i}\in M_{k}^{t}$, and $M_{k}^{r}=M_{k-1}^{c}\setminus M_{k}^{t}$. Otherwise, if no $m^{i}\in M_{k-1}^{c}$ satisfies (Equation 27), $M_{k}^{c}=M_{k-1}^{c}$.

$M_{k}^{a}$ is constructed based on Rényi entropy from $M_{k}^{fa}$. The matching degree can be measured by the Rényi entropy $D(s_{k},m_{k}^{j})$ between $m_{k}^{j}\in M_{k}^{fa}$ and $s_{k}$, where smaller values correspond to higher distributional similarity. Using $z_{k}$ as the common variable, $D(s_{k},m_{k}^{j})$ is expressed as:(28)$$\begin{array}{cc} D\left(s_{k},m_{k}^{j}\right)= & D\left[p\left(z_{k}|s_{k}\right)∥p\left(z_{k}|m_{k}^{j}\right)\right]\\ = & -\frac{1}{1-\lambda_{0}}\ln\int p^{\lambda_{0}}\left(z_{k}|M_{1:k-1}^{c},s_{k},z_{1:k-1}\right)p^{1-\lambda_{0}}\left(z_{k}|M_{1:k-1}^{c},m_{k}^{j},z_{1:k-1}\right)\ln dz_{k} \end{array}$$
where $\lambda_{0}$ is a tuning parameter. As $\lambda_{0}\to 1$, the Rényi divergence becomes the Kullback–Leibler (KL) divergence. The sequence $M_{1:k-1}^{c}$ denotes the coarse model sets from time 1 to k−1, and $z_{1:k-1}$ represents the measurement sequence up to time k−1. Under Gaussian assumptions, it follows that:(29)$$p\left(z_{k}|M_{1:k-1}^{c},s_{k},z_{1:k-1}\right)=N\left(z_{k};{\bar{z}}_{k}^{s_{k}},P_{zz,k}^{s_{k}}\right)$$(30)$$p\left(z_{k}|M_{1:k-1}^{c},m_{k}^{j},z_{1:k-1}\right)=N\left(z_{k};{\bar{z}}_{k}^{j},{\hat{P}}_{zz,k}^{j}\right)$$
$s_{k}$ is approximated by $M_{k-1}^{c}$:(31)$$p\left(z_{k}|M_{1:k-1},s_{k},z_{1:k-1}\right)\approx p\left(z_{k}|M_{1:k-1},M_{k-1}^{c},z_{1:k-1}\right)$$(32)$$\begin{array}{cc} {\bar{z}}_{k}^{s_{k}} & \approx E\left\{z_{k}|M_{1:k-1},M_{k-1}^{c},z_{1:k-1}\right\}\\ \; & =\sum_{m_{k}^{j}\in M_{k-1}^{c}}{\hat{z}}_{k|k-1}^{j}\mu_{k|k-1}^{j} \end{array}$$(33)$$\begin{array}{cc} \; & P_{zz,k}^{s_{k}}\approx E\left\{\left(z_{k}-{\bar{z}}_{k}^{s_{k}}\right){\left(z_{k}-{\bar{z}}_{k}^{s_{k}}\right)}^{T}|M_{k-1}^{c},z_{1:k-1}\right\}\\ \; & =\sum_{m_{k}^{j}\in M_{k-1}^{c}}\left(P_{zz,k|k-1}^{j}+\left(z_{k}-{\bar{z}}_{k}^{s_{k}}\right){\left(z_{k}-{\bar{z}}_{k}^{s_{k}}\right)}^{T}\right)\mu_{k|k-1}^{j} \end{array}$$
where $m_{k}^{j}\in M_{k}^{fa}$.

Substitute (Equation 29)–(Equation 33) into (Equation 28):(34)$$\begin{array}{cc} D\left(s_{k},m_{k}^{j}\right)= & -\frac{1}{2\left(1-\lambda_{0}\right)}\ln\frac{{\left|{\hat{P}}_{zz,k}^{j}\right|}^{\lambda_{0}}{\left|{\hat{P}}_{zz,k}^{s_{k}}\right|}^{1-\lambda_{0}}}{\left|\lambda_{0}P_{zz,k}^{j}+\left(1-\lambda_{0}\right)P_{zz,k}^{s_{k}}\right|}\\ \; & +\frac{\lambda_{0}}{2}{\left({\bar{z}}_{k}^{s_{k}}-{\bar{z}}_{k}^{j}\right)}^{T}{\left(\lambda_{0}{\hat{P}}_{zz,k}^{j}+\left(1-\lambda_{0}\right)P_{zz,k}^{s_{k}}\right)}^{-1}\left({\bar{z}}_{k}^{s_{k}}-{\bar{z}}_{k}^{j}\right) \end{array}$$

In $M_{k}^{a}$, the following initialization strategy is adopted for $M_{k}^{a}$ to accelerate filter convergence:(35)$${\hat{x}}_{k-1}^{j}={\hat{x}}_{k-1}^{*},\;{\hat{P}}_{k-1}^{j}={\hat{P}}_{k-1}^{*}$$
where ${\hat{x}}_{k-1}^{*}$ and ${\hat{P}}_{k-1}^{*}$ denote the state vector and covariance matrix corresponding to the model with the highest posterior probability in $M_{k-1}^{c}$.

### 3.3. Design of Fine Model Set

The fine model set improves target motion mode characterization accuracy. Its design consists of mode moment computation and fine model set construction.

For $M^{k}$, the expected motion mode is defined as:(36)$${\bar{m}}_{k}={\bar{m}}_{k}^{A_{k}}\mu_{k}^{A_{k}}+{\bar{m}}_{k}^{M_{k}^{c}}\mu_{k}^{M_{k}^{c}}$$

Specifically, ${\bar{m}}_{k}$ corresponds to the model parameters $\left(\lambda ,\omega ,\beta ,\sigma_{a}\right)$. When designing the fine model set at time *k*, the expected motion mode ${\bar{m}}_{k}^{A_{k}}$ and model probability $\mu_{k}^{A_{k}}$ at time *k* are unavailable. These values are approximated using ${\bar{m}}_{k}^{A_{k-1}}$ and $\mu_{k}^{A_{k-1}}$. Thus, ${\bar{m}}_{k}$ is computed as:(37)$$\begin{array}{cc} {\bar{m}}_{k} & \approx {\bar{m}}_{k-1}^{A_{k-1}}\mu_{k-1}^{A_{k-1}}+{\bar{m}}_{k-1}^{M_{k-1}^{c}}\mu_{k-1}^{M_{k-1}^{c}}\\ \; & =\sum_{i=1}^{n_{A_{k-1}}}{\bar{m}}_{k-1}^{r}\mu_{k-1}^{r}+\sum_{r=1}^{n_{M_{k-1}^{c}}}{\bar{m}}_{k-1}^{i}\mu_{k-1}^{i} \end{array}$$

The covariance of ${\bar{m}}_{k}$ is given by(38)$$\begin{array}{cc} \Sigma_{k} & =\sum_{r=1}^{n_{A_{k-1}}}\left[\Sigma_{k-1}^{r}+\left({\bar{m}}_{k-1}^{r}-{\bar{m}}_{k}\right){\left({\bar{m}}_{k-1}^{r}-{\bar{m}}_{k}\right)}^{T}\right]\mu_{k-1}^{r}\\ \; & +\sum_{i=1}^{n_{M_{k-1}^{c}}}\left[\Sigma^{i}+\left({\bar{m}}_{k-1}^{i}-{\bar{m}}_{k}\right){\left({\bar{m}}_{k-1}^{i}-{\bar{m}}_{k}\right)}^{T}\right]\mu_{k-1}^{i} \end{array}$$

The fine-model set ${\left\{\mu_{k|k-1}^{r|A_{k}},{\bar{m}}_{k}^{r}\right\}}_{r=1}^{n_{A_{k}}}$ at time *k* is designed by moment matching method. This ensures first two moments match ${\bar{m}}_{k}$ and $\Sigma_{k}$, respectively:(39)$${\bar{m}}_{k}=\sum_{r=1}^{n_{A_{k}}}\mu_{kk-1}^{r|A_{k}}{\bar{m}}_{k}^{r}$$(40)$$\Sigma_{k}=\sum_{r=1}^{n_{A_{k}}}\mu_{k|k-1}^{r|A_{k}}\left[\Sigma_{k}^{r}+\left({\bar{m}}_{k}^{r}-{\bar{m}}_{k}\right){\left({\bar{m}}_{k}^{r}-{\bar{m}}_{k}\right)}^{T}\right]$$

Here, ${\bar{m}}_{k}^{r}$ and $\Sigma_{k}^{r}$ denote the first-order and second-order moments of $m_{k}^{r}$, respectively. Define $\Sigma_{k}^{r}=\left(1-\rho \right)\Sigma_{k}$, (Equation 40) can be reformulated as:(41)$$\begin{array}{cc} \Sigma_{k}= & \sum_{r=1}^{n_{Ak}}\mu_{k|k-1}^{r|A_{k}}\left[\left(1-\rho \right)\Sigma_{k}\right.\left.+\left({\bar{m}}_{k}^{\left(r\right)}-{\bar{m}}_{k}\right){\left({\bar{m}}_{k}^{\left(r\right)}-{\bar{m}}_{k}\right)}^{T}\right] \end{array}$$
where 0≤ρ<1, $0\leq \mu_{k|k-1}^{r|A_{k}}<1$. (Equation 41) could be simplified as(42)$$\rho \Sigma_{k}=\sum_{r=1}^{n_{A_{k}}}\mu_{k|k-1}^{r|A_{k}}\left({\bar{m}}_{k}^{\left(r\right)}-{\bar{m}}_{k}\right){\left({\bar{m}}_{k}^{\left(r\right)}-{\bar{m}}_{k}\right)}^{T}$$

Consequently, the designed fine model set should satisfy the following conditions:(43)$$\begin{array}{cc} \; & \sum_{r=1}^{n_{A_{k}}}\mu_{k|k-1}^{r|A_{k}}=1\;\sum_{r=1}^{n_{A_{k}}}\mu_{k|k-1}^{r|A_{k}}{\bar{m}}_{k}^{\left(r\right)}={\bar{m}}_{k}\\ \; & \sum_{r=1}^{n_{A_{k}}}\mu_{k|k-1}^{r|A_{k}}\left({\bar{m}}_{k}^{\left(r\right)}-{\bar{m}}_{k}\right){\left({\bar{m}}_{k}^{\left(r\right)}-{\bar{m}}_{k}\right)}^{T}=\rho \Sigma_{k} \end{array}$$

Introduce the variable ${\bar{s}}_{k}^{r}$ subject to the following conditions:(44)$${\bar{m}}_{k}^{r}=\Lambda {\bar{s}}_{k}^{r}+{\bar{m}}_{k}$$
where Λ is obtained by Cholesky decomposition of $\rho \Sigma_{k}$, satisfying $\rho \Sigma_{k}=\Lambda \Lambda^{T}$.

The design of the model set ${\left\{\mu_{k|k-1}^{r|A_{k}},{\bar{m}}_{k}^{r}\right\}}_{r=1}^{n_{A_{k}}}$ can thus be reformulated as the design of a parameter set ${\left\{\mu_{k|k-1}^{r|A_{k}},{\bar{s}}_{k}^{r}\right\}}_{r=1}^{n_{A_{k}}}$ subject to the following conditions:(45)$$\begin{array}{cc} \; & \sum_{r=1}^{n_{A_{k}}}\mu_{k|k-1}^{r|A_{k}}=1\;\sum_{r=1}^{n_{A_{k}}}\mu_{k|k-1}^{r|A_{k}}{\bar{s}}_{k}^{r}=0\\ \; & \sum_{r=1}^{n_{A_{k}}}\mu_{k|k-1}^{r|A_{k}}{\bar{s}}_{k}^{r}{\left({\bar{s}}_{k}^{r}\right)}^{T}=I \end{array}$$

To satisfy (Equation 45) under reasonable computational complexity, we propose a geometrically symmetric design employing $n_{A_{k}}=d+2$ points. This is achieved by constructing a regular simplex in $R^{d}$ defined by d+1 vectors $u_{0},u_{1},\ldots ,u_{d}$ with the Gram matrix:(46)$$u_{i}^{T}u_{j}=\left\{\begin{array}{cc} d & i=j\\ -1 & i\neq j \end{array}\right.$$
$u_{0},u_{1},\ldots ,u_{d}$ are scaled by parameter s>d (typically s=d+1) to form the first d+1 sigma points:(47)$${\bar{s}}_{k}^{r}=\sqrt{\frac{s}{d}}u_{r-1},$$
with the ${\left(d+2\right)}^{\mathrm{th}}$ point fixed at the origin:(48)$${\bar{s}}_{k}^{\left(d+2\right)}=0$$

The corresponding weights are assigned as:(49)$$\left\{\begin{array}{c} \mu_{k|k-1}^{r|A_{k}}=\frac{d}{(d+1)s},r=1,2,\ldots d+1\\ \mu_{k|k-1}^{\left(d+2\right)|A_{k}}=1-\frac{d}{s} \end{array}\right.$$

The proof is provided in Appendix A.

### 3.4. Global Estimation Fusion

Following coarse model set adaptation, the mode-conditioned estimates ${\hat{x}}_{k}^{M_{k}^{c}}$ and ${\hat{P}}_{k}^{M_{k}^{c}}$ are utilized to compute global state estimates via the IMM algorithm:1.Interaction input:The predicted probability $\mu_{k|k-1}^{i}$ is computed by:(50)$$\mu_{k|k-1}^{i}=\sum_{j=1}^{r}\pi_{ji}\mu_{k-1}^{j},\;i=1,2,\cdots ,n_{M_{k}^{c}}$$
where $\pi_{ji}$ is the element of the Markov transition matrix Π, and $\mu_{k-1}^{j}$ denotes the probability of model *j* in the coarse model set $M_{k-1}^{c}$.The mixing weight $\mu_{k-1}^{j|i}$ is given by:(51)$$\mu_{k}^{j|i}=\frac{\pi_{ji}\mu_{k-1}^{j}}{\mu_{k|k-1}^{i}},\;i,j=1,2,\cdots ,n_{M_{k}^{c}}$$The mixed state estimate ${\bar{x}}_{k-1}^{i}$ and its covariance matrix ${\bar{P}}_{k-1}^{i}$ is computed by:(52)$${\bar{x}}_{k-1}^{i}=\sum_{j=1}^{n_{M_{k}^{c}}}\mu_{k}^{j|i}{\hat{x}}_{k-1}^{j},\;i=1,2,\cdots ,n_{M_{k}^{c}}$$(53)$$\begin{array}{cc} \; & {\bar{P}}_{k-1}^{i}=\sum_{j=1}^{n_{M_{k}^{c}}}\mu_{k-1}^{j|i}\left(P_{k-1}^{j}\right.\\ \; & \left.+\left({\hat{x}}_{k-1}^{j}-{\bar{x}}_{k-1}^{i}\right){\left({\hat{x}}_{k-1}^{j}-{\bar{x}}_{k-1}^{i}\right)}^{T}\right) \end{array}$$2.Parallel filtering and model probability update:The subfilters initialize with the mixed estimate ${\bar{x}}_{k-1}^{i}$ and mixed covariance ${\bar{P}}_{k-1}^{i}$ derived from the interaction step. For each model $m_{k}^{i}$, the time update is performed to compute the state estimate:(54)$$\begin{array}{cc} \; & \left[v_{k}^{i},S_{k}^{i},{\hat{x}}_{k}^{i},P_{k}^{i},\right]=\mathrm{filter}\left(m_{k}^{i},{\bar{x}}_{k-1}^{i},{\bar{P}}_{k-1}^{i},z_{k}\right) \end{array}$$
where filter(·) denotes the filtering function. $v_{k}^{i}$, $S_{k}^{i}$, ${\hat{x}}_{k}^{i}$, and $P_{k}^{i}$ represent the innovation vector, innovation covariance matrix, state estimate, and state error covariance matrix output by subfilter *i* at time *k*, respectively.Compute the likelihood function:(55)$$L_{k}^{i}=\frac{\exp\left[-{\left(v_{k}^{i}\right)}^{T}{\left(S_{k}^{i}\right)}^{-1}v_{k}^{i}/2\right]}{{\left|2\pi S_{k}^{i}\right|}^{1/2}},\;i=1,2,\cdots ,n_{M_{k}^{c}}$$(56)$$c_{1}=\sum_{i=1}^{n_{M_{k}^{c}}}\mu_{k|k-1}^{i|M_{k}^{c}}L_{k}^{i}$$The posterior model probability of $m_{k}^{i}$ is(57)$$\mu_{k}^{i|M_{k}^{c}}=\frac{\mu_{k|k-1}^{i|M_{k}^{c}}L_{k}^{i}}{c_{1}}$$3.Estimation fusion in coarse model set:The state estimate ${\hat{x}}_{k}^{M_{k}^{c}}$ and its covariance matrix ${\hat{P}}_{k}^{M_{k}^{c}}$ for the coarse model set at time *k* are given by:(58)$${\hat{x}}_{k}^{M_{k}^{c}}=\sum_{i=1}^{n_{M_{k}^{c}}}\mu_{k}^{i|M_{k}^{c}}{\hat{x}}_{k}^{i}$$(59)$$\begin{array}{cc} {\hat{P}}_{k}^{M_{k}^{c}} & =\sum_{i=1}^{n_{M_{k}^{c}}}\mu_{k}^{i|M_{k}^{c}}\left({\hat{P}}_{k}^{i}\right.\left.+\left({\hat{x}}_{k}^{i}-{\hat{x}}_{k}^{M_{k}^{c}}\right){\left({\hat{x}}_{k}^{i}-{\hat{x}}_{k}^{M_{k}^{c}}\right)}^{T}\right) \end{array}$$
The state estimate ${\hat{x}}_{k}^{A_{k}}$ and covariance ${\hat{P}}_{k}^{A_{k}}$ of the fine model set can be expressed as:(60)$${\hat{x}}_{k}^{A_{k}}=\sum_{r=1}^{n_{A_{k}}}\mu_{k}^{r|A_{k}}{\hat{x}}_{k}^{r},r=1,2,\cdots ,n_{A_{k}}$$(61)$$\begin{array}{cc} {\hat{P}}_{k}^{A_{k}} & =\sum_{r=1}^{n_{A_{k}}}\mu_{k}^{r|A_{k}}\left({\hat{P}}_{k}^{r}\right.\left.+\left({\hat{x}}_{k}^{r}-{\hat{x}}_{k}^{A_{k}}\right){\left({\hat{x}}_{k}^{r}-{\hat{x}}_{k}^{A_{k}}\right)}^{T}\right) \end{array}$$
where $\mu_{k}^{r|A_{k}}$ denotes the posterior probability that the fine model $m_{k}^{r}$ belongs to the fine model set $A_{k}$ at time *k*, and ${\hat{x}}_{k}^{r}$, ${\hat{P}}_{k}^{r}$ represent the state estimate and covariance conditioned on $m_{k}^{r}$ given by(62)$$\begin{array}{cc} \; & \left[{\hat{x}}_{k}^{r},{\hat{P}}_{k}^{r},v_{k}^{r},S_{k}^{r}\right]=\mathrm{filter}\left(m_{k}^{r},{\hat{x}}_{k-1}^{r},{\hat{P}}_{k-1}^{r},z_{k}\right) \end{array}$$

The posterior model probability of $m_{k}^{r}$ in fine model set is(63)$$\mu_{k}^{r|A_{k}}=\frac{1}{c_{2}}L_{k}^{r}\mu_{k|k-1}^{r|A_{k}},r=1,2,\cdots ,n_{A_{k}}$$
where(64)$$L_{k}^{r}=\frac{\exp\left[-{\left(v_{k}^{r}\right)}^{T}{\left(S_{k}^{r}\right)}^{-1}v_{k}^{r}/2\right]}{{\left|2\pi S_{k}^{r}\right|}^{1/2}},r=1,2,\cdots ,n_{A_{k}}$$(65)$$c_{2}=\sum_{r=1}^{n_{A_{k}}}L_{k}^{r}\mu_{k|k-1}^{r|A_{k}}$$

The model predicted probability $\mu_{k|k-1}^{r}$ of $m_{k}^{r}$ is essential for computing the global posterior probability $\mu_{k}^{A_{k}}$. However, due to the dynamic reconfiguration of the fine model set based on subfilter outputs, defining a conventional transition probability matrix is impractical. Consequently, this probability is computed via an alternative approach:(66)$$\begin{array}{cc} \mu_{k|k-1}^{r} & =\mathrm{Pr}\left\{m_{k}^{r}|M^{k},z^{k-1}\right\}\\ \; & =\mathrm{Pr}\left\{m_{k}^{r}|A_{k},M^{k-1},z^{k-1}\right\}\mathrm{Pr}\left\{A_{k}|M^{k},z^{k-1}\right\}\\ \; & =\mu_{k|k-1}^{r|A_{k}}\mu_{k|k-1}^{A_{k}},r=1,2,\cdots ,n_{A_{k}} \end{array}$$

Disregard discontinuous transitions between fine model subsets and coarse model sets:(67)$$\mu_{k|k-1}^{A_{k}}=\mu_{k-1}^{A_{k}}=\mathrm{Pr}\left\{A_{k-1}|M^{k-1},z^{k-1}\right\}$$
then (Equation 66) becomes(68)$$\mu_{k|k-1}^{r}=\mu_{k|k-1}^{r|A_{k}}\mu_{k-1}^{A_{k}}$$

The global posterior probability of $A_{k}$ is(69)$$\begin{array}{cc} \mu_{k}^{A_{k}}= & \frac{\sum_{r=1}^{n_{A_{k}}}\mathrm{Pr}\left\{z_{k}|m_{k}^{r},M^{k-1},z^{k-1}\right\}\mathrm{Pr}\left\{m_{k}^{r}|M^{k},z^{k-1}\right\}}{\mathrm{Pr}\left\{z_{k}|M^{k},z^{k-1}\right\}}\\ = & \frac{\sum_{r=1}^{n_{A_{k}}}L_{k}^{r}\mu_{k|k-1}^{r}}{\mathrm{Pr}\left\{z_{k}|M^{k},z^{k-1}\right\}}=\frac{\sum_{r=1}^{n_{A_{k}}}L_{k}^{r}\mu_{k|k-1}^{r|A_{k}}\mu_{k-1}^{A_{k}}}{\mathrm{Pr}\left\{z_{k}|M^{k},z^{k-1}\right\}}\\ = & \frac{c_{2}\mu_{k-1}^{A_{k-1}}}{c_{1}\mu_{k-1}^{M_{k-1}^{c}}+c_{2}\mu_{k-1}^{A_{k}}} \end{array}$$
where $\mu_{k-1}^{M_{k-1}^{c}}$ represents the global posterior probability of the coarse model set $M_{k-1}^{c}$ at time step k−1. The probability of the coarse model subset $M_{k}^{c}$ at time step *k* is given by:(70)$$\mu_{k}^{M_{k}^{c}}=1-\mu_{k}^{A_{k}}=\frac{c_{1}\mu_{k-1}^{M_{k}^{c}}}{c_{1}\mu_{k-1}^{M_{k}^{c}}+c_{2}\mu_{k-1}^{A_{k}}}$$

The global state estimate ${\hat{x}}_{k}$ and its covariance matrix ${\hat{P}}_{k}$ are given by:(71)$${\hat{x}}_{k}=\mu_{k}^{M_{k}^{c}}{\hat{x}}_{k}^{M_{k}^{c}}+\mu_{k}^{A_{k}}{\hat{x}}_{k}^{A_{k}}$$(72)$${\hat{P}}_{k}=\mu_{k}^{M_{k}^{c}}{\hat{P}}_{k}^{M_{k}^{c}}+\mu_{k}^{A_{k}}{\hat{P}}_{k}^{A_{k}}$$
The overall procedure of the HAMM multi-model approach is summarized in Table 2.

## 4. Design of Robust Filter

The tracking model set is constructed to describe HGT’s motion characteristics accurately, and the HAMM method is designed to address the problem of model parameters, but the filtering process is still disrupted by non-Gaussian noise, and a robust filter with high stability is required. Therefore, in this section, MEECKF is proposed by constructing a regression model and implementing robust state estimation, respectively.

### 4.1. Regression Model

The regression model is utilized for transitioning the state estimation to the implementation of the MEE criterion. Given the nonlinearity of (Equation 16), the statistical linearization method is adopted to precisely approximate the nonlinear function for constructing the regression model through(73)$$z_{k}={\hat{z}}_{k|k-1}+H_{k}^{*}\left(x_{k}-{\hat{x}}_{k|k-1}\right)+v_{k}$$
where ${\hat{z}}_{k|k-1}$ is the measurement prediction vector. $H_{k}^{*}$ is the pseudo-measurement matrix and it is calculated by(74)$$H_{k}^{*}={\left({\hat{P}}_{k|k-1}^{-1}{\hat{P}}_{xz}\right)}^{T}$$
where ${\hat{P}}_{xz}$ is the cross covariance between ${\hat{x}}_{k|k-1}$ and ${\hat{z}}_{k|k-1}$.

The prediction covariance matrix ${\hat{P}}_{k|k-1}$ and ${\hat{P}}_{xz}$ can be computed in the process of CKF. However, the standard CKF employs Cholesky decomposition to generate sample points, necessitating the error covariance matrix to maintain strict positive definiteness. Due to inevitable errors induced by measurement noise under the plasma sheath, the filtering process may compromise this positive condition of the covariance matrix over successive iterations. To enhance algorithmic stability, singular value decomposition is utilized for sample point calculation instead of Cholesky decomposition. $H_{k}^{*}$ can be obtained, and the regression model can be constructed with such an improvement. Therefore, ${\hat{P}}_{k|k-1}$ and ${\hat{P}}_{xz}$ are calculated through(75)$${\hat{P}}_{k-1}=U_{k-1}S_{k-1}V_{k-1}^{T}$$(76)$$x_{k-1}^{i}={\hat{x}}_{k-1}+U_{k-1}\sqrt{S_{k-1}}\xi^{\left(i\right)}$$(77)$$x_{k|k-1}^{i}=f\left(x_{k-1}^{i}\right)$$(78)$${\hat{x}}_{k|k-1}=\sum_{i=1}^{m}\mu_{i}x_{k|k-1}^{i}$$(79)$$\begin{array}{cc} {\hat{P}}_{k|k-1}= & \sum_{i=1}^{m}\mu_{i}\left(x_{k|k-1}^{i}-{\hat{x}}_{k|k-1}\right){\left(x_{k|k-1}^{i}-{\hat{x}}_{k|k-1}\right)}^{T}+Q_{x,k} \end{array}$$(80)$${\hat{P}}_{k|k-1}=U_{k|k-1}S_{k|k-1}V_{k|k-1}^{T}$$(81)$$x_{k|k-1}^{i}={\hat{x}}_{k|k-1}+U_{k|k-1}\sqrt{S_{k|k-1}}\xi^{\left(i\right)}$$(82)$$z_{k|k-1}^{i}=h\left(x_{k|k-1}^{i}\right)$$(83)$${\hat{z}}_{k|k-1}=\sum_{i=1}^{m}\mu_{i}z_{k|k-1}^{i}$$(84)$$\begin{array}{cc} {\hat{P}}_{zz}= & \sum_{i=1}^{m}\mu_{i}\left(z_{k|k-1}^{i}-{\hat{z}}_{k|k-1}\right){\left(z_{k|k-1}^{i}-{\hat{z}}_{k|k-1}\right)}^{T}+R \end{array}$$(85)$${\hat{P}}_{xz}=\sum_{i=1}^{m}\mu_{i}\left(x_{k|k-1}^{i}-{\hat{x}}_{k|k-1}\right){\left(z_{k|k-1}^{i}-{\hat{z}}_{k|k-1}\right)}^{T}$$
where $\xi^{\left(i\right)}=\sqrt{\frac{1}{m}}{\left[1\right]}_{i},\mu_{i}=\frac{1}{m},i=1,2,\ldots m,m=2n$, *n* is the dimension of state vector.

The regression model can be constructed by(86)$$D_{k}=W_{k}x_{k}+e_{k}$$
where(87)$$D_{k}=\Theta_{k}^{-1}\left[\begin{array}{c} {\hat{x}}_{k|k-1}\\ z_{k}^{*}-{\hat{z}}_{k|k-1}^{*}+H_{k}^{*}{\hat{x}}_{k|k-1} \end{array}\right]$$(88)$$W_{k}=\Theta_{k}^{-1}\left[\begin{array}{c} I_{n}\\ H_{k}^{*} \end{array}\right]$$(89)$$e_{k}=\Theta_{k}^{-1}\left[\begin{array}{c} -\left(x_{k}-{\hat{x}}_{k|k-1}\right)\\ \zeta_{k-1} \end{array}\right]$$
$I_{n}$ is identity matrix, $\Theta_{k}$ is obtained by(90)$$\begin{array}{cc} \Theta_{k} & =\left[\begin{array}{cc} \Theta_{p} & 0\\ 0 & \Theta_{r} \end{array}\right]=\mathrm{chol}\left[\begin{array}{cc} {\hat{P}}_{k|k-1} & 0\\ 0 & R_{m} \end{array}\right] \end{array}$$

### 4.2. Robust State Estimation

The proposed filter performs state estimation by performing the MEE criterion over the constructed regression model in (Equation 86). The definition and detailed derivation of the error entropy can be found in [28]. For brevity, we directly present the cost function(91)$$J_{l_{e}}\left(x_{k}\right)=\frac{1}{l_{e}^{2}}\sum_{i=1}^{l_{e}}\sum_{j=1}^{l_{e}}G_{\sigma_{k}}\left(e_{j,k}-e_{i,k}\right)$$
where $l_{e}$ denotes the dimension number of $e_{k}$. $e_{j,k}$ is the *j*-th row of the error vector $e_{k}$, and $G_{\sigma_{k}}\left(a\right)=\exp\left(-a^{2}/2{\sigma_{k}}^{2}\right)$ is the kernel function with the width $\sigma_{k}$. $\sigma_{k}$ directly affects MEE criterion’s suppression of outliers. Smaller values result in stronger suppression but may lead to filter divergence.

State estimate is obtained by maximizing the cost function by(92)$${\hat{x}}_{k}=\arg\max_{x_{k}}J_{l}\left(x_{k}\right).$$

By equating the gradient of the cost function with respect to $x_{k}$ to zero, we obtain(93)$$W_{k}^{T}\Psi_{k}e_{k}=W_{k}^{T}\Phi_{k}e_{k}$$
where(94)$${\left[\Phi_{k}\right]}_{ij}=G_{\sigma_{k}}\left(e_{j,k}-e_{i,k}\right)$$(95)$$\Psi_{k}=\mathrm{diag}\left\{\sum_{i=1}^{L}G_{\sigma_{k}}\left(e_{1,k}-e_{i,k}\right),\ldots \sum_{i=1}^{L}G_{\sigma_{k}}\left(e_{L,k}-e_{i,k}\right)\right\}.$$

To obtain the state estimation vector ${\hat{x}}_{k}$, (Equation 92) is solved through the fixed-point iteration method, and the iteration equation is expressed as(96)$${\hat{x}}_{k}=g\left({\hat{x}}_{k}\right)={\left(W_{k}^{T}\Lambda_{k}W_{k}\right)}^{-1}\left(W_{k}^{T}\Lambda_{k}D_{k}\right)$$
where(97)$$\Lambda_{k}=\Psi_{k}-\Phi_{k}.$$

However, it is highly probable that (Equation 96) may diverge during the filtering process. When non-Gaussian measurement noise is present, large outliers in measurements are likely to occur, which results in significantly large errors $e_{k}$. According to the kernel function $G_{\sigma_{k}}\left(\cdot \right)$, the element of matrix $\Lambda_{k}$ tend towards zero, leading to singularity in matrix $\Lambda_{k}$. This singularity causes instability in the numerical computation of the inverse of matrix ${W_{k}}^{T}\Lambda_{k}W_{k}$.

Setting the gradient of the cost function to zero yields an alternative expression to (Equation 96), which is expressed as(98)$${\left[\Psi_{k}^{T},\Phi_{k}^{T}\right]}^{T}W_{k}x_{k}={\left[\Psi_{k}^{T},\Phi_{k}^{T}\right]}^{T}D_{k}$$

The state estimation is given by(99)$$\left\{\begin{array}{c} {\hat{x}}_{k}={\left[W_{k}^{T}\Omega_{k}W_{k}\right]}^{-1}W_{k}^{T}\Omega_{k}D_{k}\\ \Omega_{k}=\Psi_{k}^{T}\Psi_{k}+\Phi_{k}^{T}\Phi_{k}. \end{array}\right.$$

The matrix $\Psi_{k}^{T}\Psi_{k}$ is definite positive, while the symmetric matrix $\Phi_{k}^{T}\Phi_{k}$ is deemed positive semidefinite. Consequently, the matrix $\Omega_{k}$ is established as positive definite, ensuring its invertibility and thereby robust state estimation.

## 5. Simulation and Results

### 5.1. Simulation Scheme

The Lockheed Martin Common Aero Vehicle (CAV) represents one of the most extensively studied hypersonic glide vehicles in current research. Ref. [43] provides the fundamental parameters and aerodynamic coefficient database for the CAV configuration. Based on the analysis of the measured data, three simulation scenarios are generated through the CAV model and the HGTs dynamics Equation (Equation 1);

Trajectories and corresponding directional accelerations are illustrated in Figure 3. Three scenarios are designed: (i) quasi-equilibrium glide trajectory with radar located at $\left[50^{\circ },60^{\circ },0\;m\right]$, (ii) jump glide trajectory with radar at $\left[29.5^{\circ },36^{\circ },0\;m\right]$, and (iii) penetration maneuver trajectory with radar at $\left[50^{\circ },60^{\circ },0\;m\right]$. Parameters include a maximum radar detection range of 600km, radar line-of-sight coverage indicated by red trajectory segments, and a sampling period T=0.1s.

To validate the effectiveness of the proposed method, we compare the proposed method against IMM, BMA, and HAMM-CKF.

The model parameter settings are first specified. The proposed HAMM method employs the model set listed in Section 2.2, with parameters given as $\alpha_{i,l}=0.1$, $\alpha_{i,s}=0.001$, $\beta_{i,l}=2\pi /10^{2}$, $\beta_{i,s}=2\pi /10^{6}$, $\sigma_{a,s}=10$ and $\sigma_{a,l}=400$, $\sigma_{\gamma ,s}=10^{-4}$ and $\sigma_{\gamma ,l}=10^{-6}$, $\sigma_{\psi ,s}=10^{-4}$ and $\sigma_{\psi ,l}=10^{-6}$. The number of coarse models is 4, and the probability transition matrix between coarse models is given by(100)$$\Pi =\left[\begin{array}{cccc} 0.9 & 0.0333 & 0.0333 & 0.0333\\ 0.0333 & 0.9 & 0.0333 & 0.0333\\ 0.0333 & 0.0333 & 0.9 & 0.0333\\ 0.0333 & 0.0333 & 0.0333 & 0.9 \end{array}\right]$$

The initial coarse model probabilities $\mu_{0}^{i|M_{0}^{c}}=0.25,\;i=1,2,3,4$, with a global probability $\mu_{0}^{M_{0}^{c}}=0.5$. The posterior probability threshold $\mu_{0}=0.1$, the frame threshold is $\tau_{f}=3$, and the fine model parameters are ρ=0.4 and $p_{0}=0.3$. The model set configuration of the proposed method remains identical across all three scenarios.

For the IMM method, the model set specified in Section 2.2 is employed, and its probability transition matrix is given by:(101)$$\Pi_{IMM}={\left[\begin{array}{ccc} 0.8 & \cdots & 0.0083\\ \vdots & \ddots & \vdots \\ 0.0083 & \cdots & 0.8 \end{array}\right]}_{25\times 25}$$

For the BMA method, the model set specified in Section 2.2 is used as the total model set, with four models participating in the estimation at each time step.

To validate the effectiveness of the proposed method, simulation experiments are conducted under different measurement noise environments, including Gaussian noise, heavy-tailed noise, and impulsive noise.

For Gaussian noise, the distribution is given by(102)$$v_{k}∼N\left(0,R\right),$$
where(103)$$R=\left[\begin{array}{ccc} \sigma_{\varphi_{a}}^{2} & 0 & 0\\ 0 & \sigma_{\theta_{e}}^{2} & 0\\ 0 & 0 & \sigma_{\rho_{r}}^{2} \end{array}\right],$$
and $\sigma_{\varphi_{a}}=0.05^{\circ }$, $\sigma_{\theta_{e}}=0.05^{\circ }$, and $\sigma_{\rho_{r}}=30\;m$ represent the standard deviations of the Gaussian noise.

For heavy-tailed noise, the measurement noise with non-Gaussian statistical characteristics is modeled as(104)$$v_{k}∼\left(1-p\right)\;N\left(0,R\right)+p\;N\left(0,R_{L}\right),$$
where p=0.1 and $R_{L}=10R$.

For impulsive noise, the measurement noise is expressed as(105)$$v_{k}∼\left(1-p\right)\;N\left(0,R\right)+p\;N\left(0,R_{L}\right),$$
where p=0.01 and $R_{L}=100R$.

In addition, to evaluate the performance of the algorithm under varying sampling rates, simulations are conducted under time-varying sampling intervals and Gaussian measurement noise conditions. For the quasi-equilibrium glide trajectory, the sampling interval is set to 0.1 s during the first 100 s and 0.5 s thereafter. For the jump glide trajectory, the sampling interval is 0.1 s during the first 60 s and 0.5 s thereafter. Similarly, for the penetration maneuver trajectory, the sampling interval is 0.1 s during the first 60 s and 0.5 s thereafter.

The estimation accuracy of all methods decreases when the sampling interval increases, reflecting the negative impact of reduced measurement frequency on state estimation performance. However, the HAMM-based methods, particularly HAMM-MEE, maintain the lowest RMSE values in position, velocity, and acceleration across all three trajectories.

In the quasi-equilibrium and jump glide phases, the transition from dense (0.1 s) to sparse (0.5 s) sampling results in a noticeable increase in RMSE for all methods. Nevertheless, the HAMM-MEE method exhibits a more gradual degradation trend, indicating better adaptability to irregular sampling conditions. In contrast, the IMM and BMA methods show larger error spikes after the sampling rate reduction, suggesting their limited robustness to asynchronous or infrequent measurements.

For the penetration maneuver, where dynamics are more abrupt, all methods suffer from higher estimation errors after the sampling interval changes. Even so, HAMM-MEE still provides the most stable performance, while HAMM-CKF performs slightly worse but remains superior to IMM and BMA. Overall, these results verify that the proposed HAMM-MEE algorithm can effectively maintain estimation accuracy under varying sampling rates, demonstrating its robustness to time-varying measurement frequencies.

In the proposed filter, the width $\sigma_{k}=4$ and the number of iterations N=15.

The simulation results are evaluated using the root mean square error (RMSE) and the average root mean square error (ARMSE):(106)$${\mathrm{RMSE}}_{k}=\sqrt{\frac{1}{L}\sum_{c=1}^{L}{\left(x_{k,c}-{\hat{x}}_{k,c}\right)}^{2}}$$(107)$$\mathrm{ARMSE}=\frac{1}{K}\sum_{k=1}^{K}{\mathrm{RMSE}}_{k}$$
where k=1,2,…,K denotes the time step index, and l=1,2,…,L represents the Monte Carlo run index. To ensure consistency of results, 200 Monte Carlo runs are conducted for each method.

The simulations were conducted on a computing platform equipped with an Intel^®^ Core™ i7-10700 CPU @ 2.90 GHz (Intel Corporation, Santa Clara, CA, USA), running MATLAB R2023b.

### 5.2. Simulation Results

#### 5.2.1. Result Under Gaussian Noise

Figure 4 presents the RMSE comparison under Gaussian noise for three trajectories: quasi-equilibrium glide, jump glide, and penetration maneuver. Each row corresponds to one trajectory, and the columns show the position, velocity, and acceleration RMSE, respectively.

Overall, HAMM-MEE and HAMM-CKF demonstrate comparable and superior performance across all tracking scenarios. Both methods exhibit rapid convergence during the initial phase and maintain stable, low RMSE values throughout the tracking process. In contrast, IMM shows significantly higher RMSE. BMA achieves intermediate performance, outperforming IMM but remaining inferior to the HAMM approaches. Under Gaussian noise conditions, HAMM-MEE and HAMM-CKF perform similarly, as theoretically expected. Their superior accuracy stems primarily from the improved multi-model adaptive mechanism rather than the estimation criterion itself. The HAMM framework more effectively handles model transitions and maintains tracking stability, particularly during steady-state phases. This advantage is consistently demonstrated across position, velocity, and acceleration estimation in all three trajectory scenarios.

Table 3 presents the ARMSE comparison of four methods under Gaussian noise across three scenarios: quasi-equilibrium glide, jump glide, and penetration maneuver. Both HAMM-based methods (HAMM-MEE and HAMM-CKF) achieve significantly lower ARMSE values in position, velocity, and acceleration compared to the IMM and BMA methods across all three scenarios. This indicates their superior estimation accuracy and robustness under Gaussian noise.

Table 4 lists the average per-time-step execution time per filtering cycle for five methods. The results demonstrate that HAMM-CKF achieves the highest computational efficiency among all methods, with an average execution time of 0.0102 s per cycle significantly lower than other approaches. BMA and HAMM-MEE exhibit moderate computational loads, requiring 0.0258 s and 0.0203 s per cycle, respectively. IMM shows substantially longer average execution times of 0.1025 s per cycle.

The simulation results under Gaussian noise demonstrate the effectiveness of the proposed HAMM method. By adaptively selecting the coarse model set and generating the fine model set, HAMM is able to better capture the target motion modes.

While the IMM approach employs multiple models to cover potential target motion modes, its extensive model set induces significant model competition, thereby degrading overall tracking performance. Comparatively, the BMA method improves model adaptability through candidate model sets, outperforming IMM. However, BMA’s continuous model set updates cause frequent replacement of active filtering models, resulting in insufficient convergence and compromised stability.

When compared to HAMM-CKF with fixed noise covariance matrices, the proposed method exhibits robustness under non-Gaussian measurement noise conditions. This advantage originates from MEE cret, which enables online adaptive estimation of noise covariance matrices, thereby enhancing target state estimation robustness.

#### 5.2.2. Result Under Heavy-Tailed Noise

Figure 5 presents the RMSE comparison under heavy-tailed noise for three trajectories: quasi-equilibrium glide, jump glide, and penetration maneuver. Each row corresponds to a trajectory, and the columns show the position, velocity, and acceleration RMSE, respectively. Overall, the proposed method demonstrates superior tracking accuracy and stability across all three tracking phases. During the initial phase, the RMSE of all methods decreases rapidly, with the proposed method and HAMM-CKF exhibiting the fastest convergence speed. Following filter convergence, the proposed method maintains low RMSE with minimal fluctuations, outperforming the traditional IMM method. During the penetration maneuver scenario at 80s, an abrupt change in target acceleration occurs, causing RMSE increases in all methods. Nevertheless, the proposed method achieves the fastest convergence speed. In summary, the proposed method surpasses the comparative algorithms (IMM, BMA) in convergence speed, steady-state accuracy, and robustness against disturbances.

Table 5 presents the ARMSE comparison of four methods under heavy-tailed noise across three flight scenarios: quasi-equilibrium glide, jump glide, and penetration maneuver. All methods experience performance degradation under heavy-tailed noise compared with the Gaussian noise case, as reflected by the increased ARMSE values in position, velocity, and acceleration. Nevertheless, the HAMM-MEE method consistently achieves the lowest errors across all three scenarios, demonstrating its strong robustness to non-Gaussian disturbances.

#### 5.2.3. Result Under Impulsive Noise

Figure 6 presents the RMSE comparison under impulsive noise for three trajectories: quasi-equilibrium glide, jump glide, and penetration maneuver. Each row corresponds to a trajectory, and the columns show the position, velocity, and acceleration RMSE, respectively. The impulsive noise environment leads to a noticeable increase in RMSE for all methods compared with the Gaussian cases, reflecting the strong disturbance effects of impulsive outliers. Nevertheless, the proposed HAMM-based methods still maintain relatively low RMSE values across all trajectories and state variables.

In particular, HAMM-MEE achieves the smallest RMSE in position, velocity, and acceleration estimation in nearly all scenarios, showing remarkable robustness to impulsive noise. HAMM-CKF also performs well but is slightly inferior to HAMM-MEE when large outliers occur.

Table 6 presents the ARMSE comparison of four methods under impulsive noise across three flight scenarios: quasi-equilibrium glide, jump glide, and penetration maneuver. The presence of impulsive noise leads to a further increase in ARMSE compared with the Gaussian and heavy-tailed noise conditions, reflecting the severe impact of outliers on state estimation accuracy. Despite this degradation, the HAMM-based methods, especially HAMM-MEE, consistently achieve lower ARMSE values than the IMM and BMA approaches across all flight scenarios.

In particular, the HAMM-MEE method demonstrates the best robustness and adaptability under impulsive disturbances, maintaining relatively stable estimation accuracy for position, velocity, and acceleration.

#### 5.2.4. Result Under Varying Sampling Rates

Figure 7 presents the RMSE comparison under varying sampling rates for three trajectories: quasi-equilibrium glide, jump glide, and penetration maneuver. Each row corresponds to a trajectory, and the columns show the position, velocity, and acceleration RMSE, respectively.

Table 7 presents the ARMSE comparison of four methods under impulsive noise across three flight scenarios: quasi-equilibrium glide, jump glide, and penetration maneuver. All methods experience a moderate increase in ARMSE under varying sampling rates compared to the fixed-rate Gaussian noise condition, indicating that reduced and irregular measurement frequency slightly degrades estimation performance. However, both HAMM-based methods (HAMM-MEE and HAMM-CKF) still achieve the lowest ARMSE values across all three flight scenarios and all state variables.

## 6. Conclusions

This paper develops a HAMM multiple-model method and MEECKF for HGTs tracking. Based on model-independent assumptions, a model set covering maneuver modes from quasi-equilibrium/jump glide to penetration maneuvers is constructed. An adaptive coarse model mechanism dynamically adjusts model combinations using Rényi divergence and posterior probability evolution, while a moment-matching fine model construction method enhances motion feature matching. Furthermore, MEECKF is proposed by utilizing singular value decomposition and adopting the MEE criterion to mitigate non-Gaussian measurement noise and enhance filtering stability. Extensive simulations validate the effectiveness of the proposed approach, demonstrating statistically significant improvements in tracking accuracy and convergence speed. The performance gains stem from both the HAMM variable-structure framework and MEE-based noise handling, as confirmed by comparative experiments under Gaussian and non-Gaussian (heavy-tailed and impulsive) noise conditions. Computational analysis shows that the method achieves superior performance with lower cost than full model-set IMM and comparable cost to BMA, making it suitable for real-time applications. Future research may investigate parallel computing architectures, robust data association strategies for varying SNR, and missed detections.

## Figures and Tables

**Figure 1 sensors-25-06621-f001:**
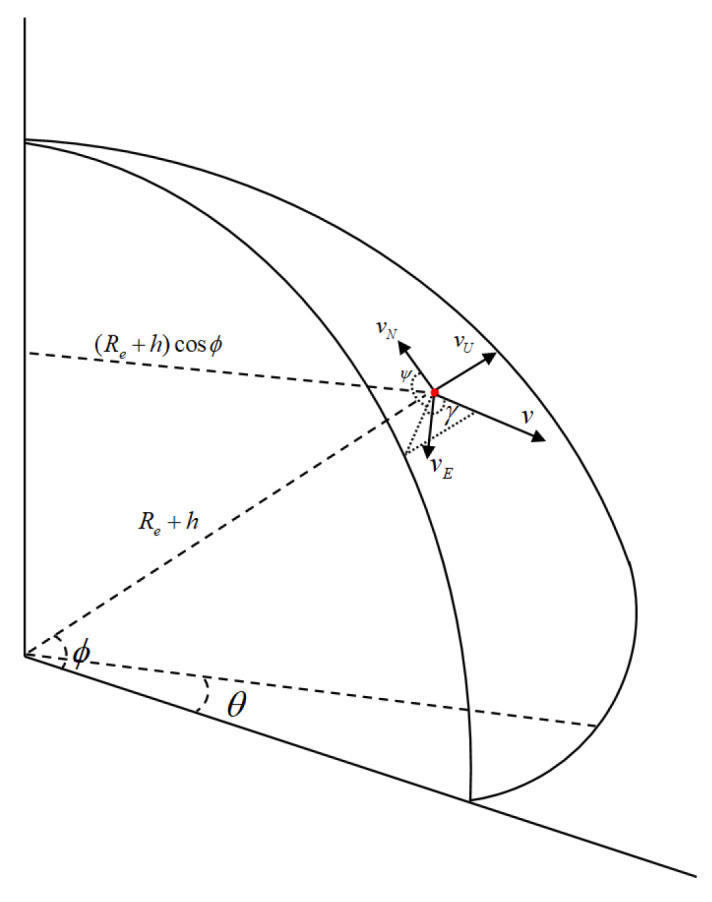
Motion model.

**Figure 2 sensors-25-06621-f002:**
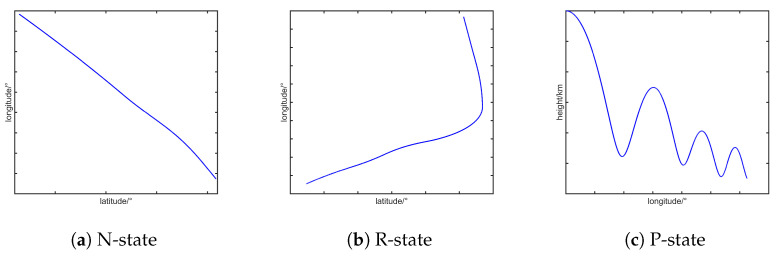
Representative trajectories for the N, R, and P maneuvering states.

**Figure 3 sensors-25-06621-f003:**
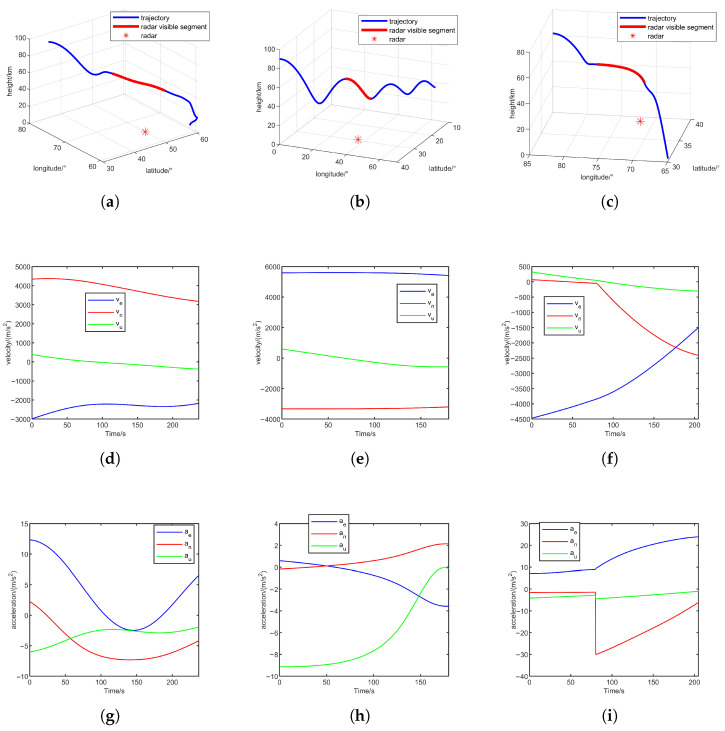
Trajectories and corresponding velocity and acceleration. (**a**) Quasi-equilibrium glide; (**b**) jump glide; (**c**) penetration maneuver; (**d**) velocity of quasi-equilibrium glide; (**e**) velocity of jump glide; (**f**) velocity of penetration maneuver; (**g**) acceleration of quasi-equilibrium glide; (**h**) acceleration of jump glide; and (**i**) acceleration of penetration maneuver.

**Figure 4 sensors-25-06621-f004:**
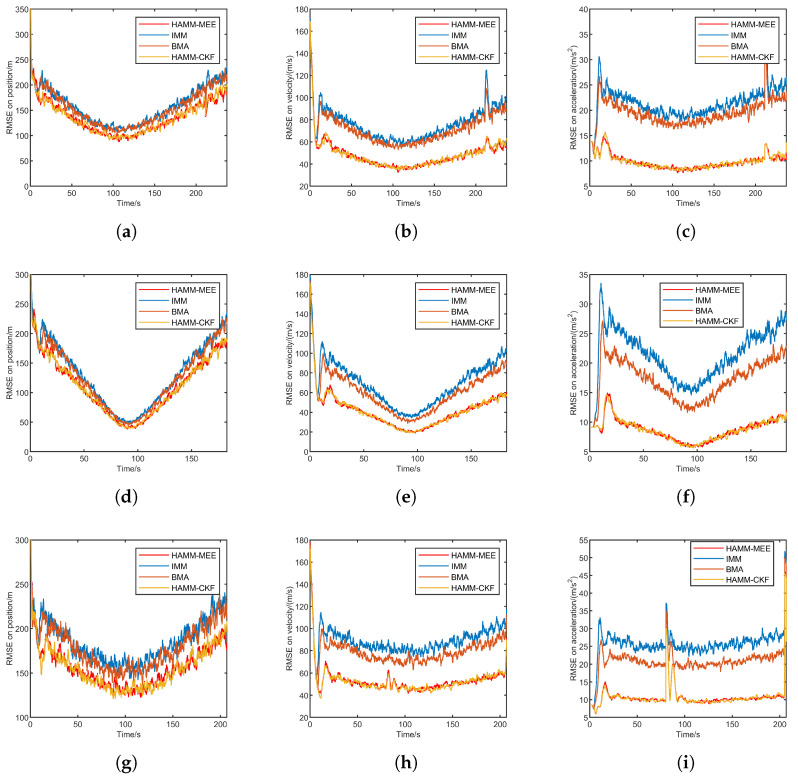
RMSE comparison under Gaussian noise for three trajectories. (**a**) Quasi-equilibrium: position. (**b**) Quasi-equilibrium: velocity. (**c**) Quasi-equilibrium: acceleration. (**d**) Jump glide: position. (**e**) Jump glide: velocity. (**f**) Jump glide: acceleration. (**g**) Penetration maneuver: position. (**h**) Penetration maneuver: velocity. (**i**) Penetration maneuver: acceleration.

**Figure 5 sensors-25-06621-f005:**
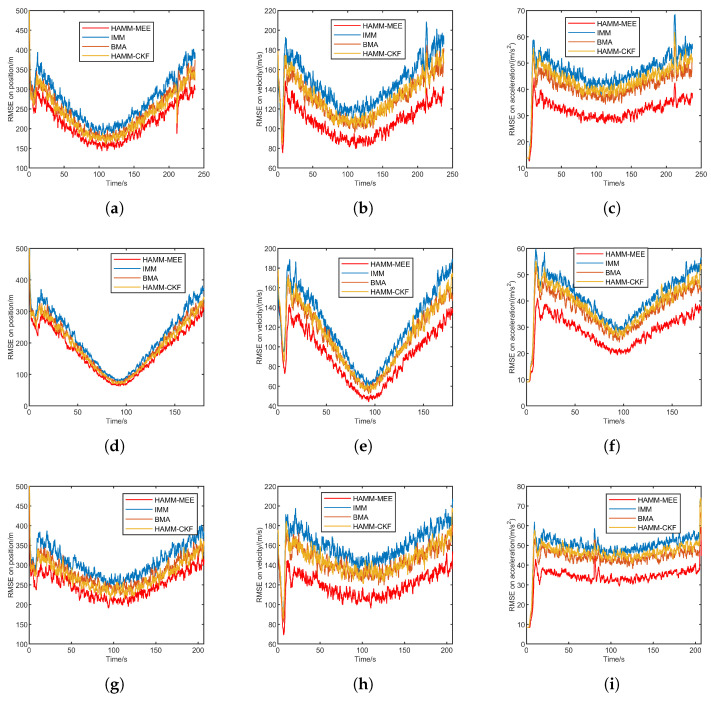
RMSE comparison under heavy-tailed noise for three trajectories. (**a**) Quasi-equilibrium: position. (**b**) Quasi-equilibrium: velocity. (**c**) Quasi-equilibrium: acceleration. (**d**) Jump glide: position. (**e**) Jump glide: velocity. (**f**) Jump glide: acceleration. (**g**) Penetration maneuver: position. (**h**) Penetration maneuver: velocity. (**i**) Penetration maneuver: acceleration.

**Figure 6 sensors-25-06621-f006:**
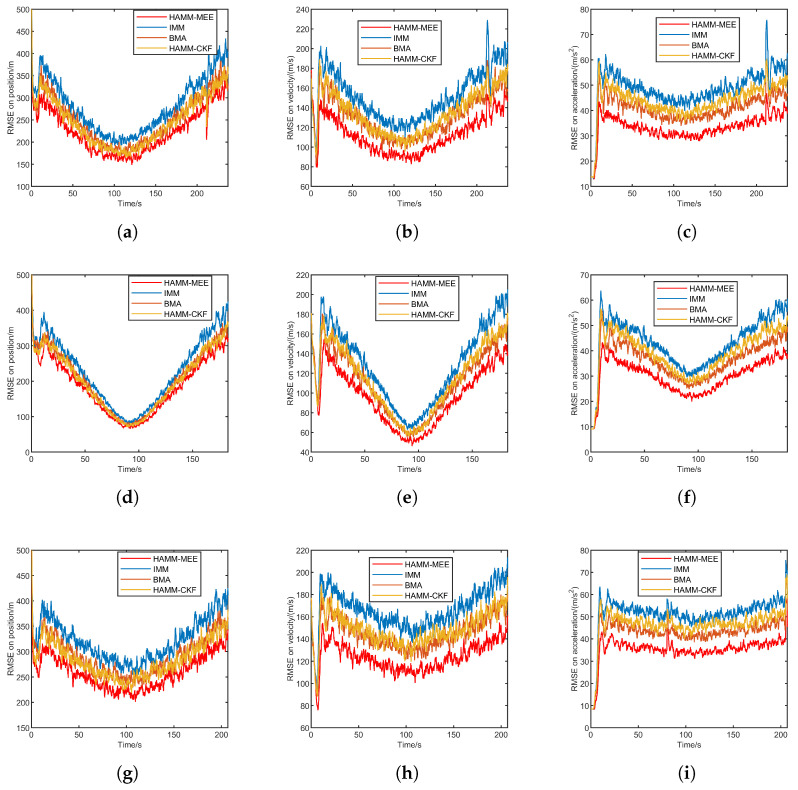
RMSE comparison under impulsive noise for three trajectories. (**a**) Quasi-equilibrium: position. (**b**) Quasi-equilibrium: velocity. (**c**) Quasi-equilibrium: acceleration. (**d**) Jump glide: position. (**e**) Jump glide: velocity. (**f**) Jump glide: acceleration. (**g**) Penetration maneuver: position. (**h**) Penetration maneuver: velocity. (**i**) Penetration maneuver: acceleration.

**Figure 7 sensors-25-06621-f007:**
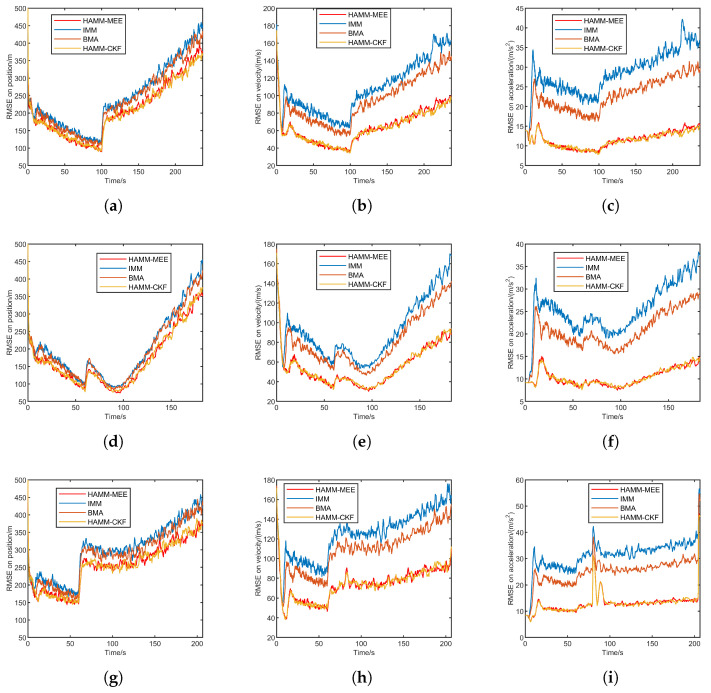
RMSE comparison under varying sampling rates for three trajectories. (**a**) Quasi-equilibrium: position. (**b**) Quasi-equilibrium: velocity. (**c**) Quasi-equilibrium: acceleration. (**d**) Jump glide: position. (**e**) Jump glide: velocity. (**f**) Jump glide: acceleration. (**g**) Penetration maneuver: position. (**h**) Penetration maneuver: velocity. (**i**) Penetration maneuver: acceleration.

**Table 1 sensors-25-06621-t001:** Parametric characterization of maneuver.

Abbreviation	Full Term	$\alpha_{i}$	$\beta_{i}$	$\sigma_{i}$
N	No maneuver	$\alpha_{i,l}$	$\beta_{i,s}$	$\sigma_{i,s}$
R	Random maneuver	$\alpha_{i,s}$	$\beta_{i,s}$	$\sigma_{i,l}$
P	Periodic maneuver	$\alpha_{i,s}$	$\beta_{i,l}$	$\sigma_{i,l}$

**Table 2 sensors-25-06621-t002:** Pseudocode summary of the proposed HAMM algorithm.

Step	Module	Operation (Key Equations and Outputs)
Initialization		
I1	Sets and priors	Init coarse set $M_{1}^{c}\subseteq M_{t}^{c}$, fine set $A_{1}$, model priors; set thresholds $\left(\mu_{0},\tau_{f}\right)$ and Rényi parameter $\lambda_{0}$. Init states/covariances for active models.
For k=1,2,…		
C1	Coarse termination	For $m^{i}\in M_{k-1}^{c}$ with activation at $k_{0}$, if $\mu_{i}^{\left(k^{\prime }\right)}<\mu_{0}$ for $k^{\prime }\in \left\{k_{0},\ldots ,k_{0}+\tau_{f}-1\right\}$ [(Equation 27)], then $m^{i}\in M_{k}^{t}$; set $M_{k}^{r}=M_{k-1}^{c}\setminus M_{k}^{t}$. If none, keep $M_{k}^{c}=M_{k-1}^{c}$.
C2	Coarse activation	Evaluate candidates $m_{k}^{j}\in M_{k}^{fa}$ by Rényi divergence $D(s_{k},m_{k}^{j})$ per (Equation 28), using Gaussian forms (Equation 29)–(Equation 33). Select small-*D* models to form $M_{k}^{a}$ and update $M_{k}^{c}=M_{k}^{r}\cup M_{k}^{a}$. Initialize new models with ${\hat{x}}_{k-1}^{j}={\hat{x}}_{k-1}^{*}$, ${\hat{P}}_{k-1}^{j}={\hat{P}}_{k-1}^{*}$.
C3	IMM interaction	Compute $\mu_{k|k-1}^{i}=\sum_{j}\pi_{ji}\mu_{k-1}^{j}$, $\mu_{k}^{j|i}=\pi_{ji}\mu_{k-1}^{j}/\mu_{k|k-1}^{i}$ by (Equation 51); form mixed inputs ${\bar{x}}_{k-1}^{i}$ by (Equation 52) and ${\bar{P}}_{k-1}^{i}$ by (Equation 53).
C4	Parallel filtering & likelihood	For each $m_{k}^{i}\in M_{k}^{c}$, run $\mathrm{filter}(m_{k}^{i},{\bar{x}}_{k-1}^{i},{\bar{P}}_{k-1}^{i},z_{k})$ to get $v_{k}^{i},S_{k}^{i},{\hat{x}}_{k}^{i},P_{k}^{i}$; compute $L_{k}^{i}$ and $c_{1}=\sum_{i}\mu_{k|k-1}^{i|M_{k}^{c}}L_{k}^{i}$; update $\mu_{k}^{i|M_{k}^{c}}=\mu_{k|k-1}^{i|M_{k}^{c}}L_{k}^{i}/c_{1}$ by (Equation 57).
C5	Fusion in $M_{k}^{c}$	${\hat{x}}_{k}^{M_{k}^{c}}=\sum_{i}\mu_{k}^{i|M_{k}^{c}}{\hat{x}}_{k}^{i}$, ${\hat{P}}_{k}^{M_{k}^{c}}=\sum_{i}\mu_{k}^{i|M_{k}^{c}}\left({\hat{P}}_{k}^{i}+\left({\hat{x}}_{k}^{i}-{\hat{x}}_{k}^{M_{k}^{c}}\right){\left({\hat{x}}_{k}^{i}-{\hat{x}}_{k}^{M_{k}^{c}}\right)}^{⊤}\right)$.
F1	Fine set moments	${\bar{m}}_{k}\approx {\bar{m}}_{k-1}^{A_{k-1}}\mu_{k-1}^{A_{k-1}}+{\bar{m}}_{k-1}^{M_{k-1}^{c}}\mu_{k-1}^{M_{k-1}^{c}}$; compute $\Sigma_{k}$ by weighted second-moment aggregation.
F2	Fine set design (matching)	Design $\left\{\left(\mu_{k|k-1}^{r|A_{k}},{\bar{m}}_{k}^{r},\Sigma_{k}^{r}\right)\right\}$ to match first two moments: (Equation 40); set $\Sigma_{k}^{r}=\left(1-\rho \right)\Sigma_{k}$ to obtain (Equation 41).
F3	Weighted simplex points	Let ${\bar{m}}_{k}^{r}=\Lambda {\bar{s}}_{k}^{r}+{\bar{m}}_{k}$, $\rho \Sigma_{k}=\Lambda \Lambda^{⊤}$, enforce (Equation 45). Use $n_{A_{k}}=d+2$ regular-simplex points with weights as specified.
F4	Fine set filtering/posteriors	Run subfilters for $m_{k}^{r}\in A_{k}$ to get ${\hat{x}}_{k}^{r},{\hat{P}}_{k}^{r},v_{k}^{r},S_{k}^{r}$; compute $L_{k}^{r}$, $c_{2}=\sum_{r}L_{k}^{r}\mu_{k|k-1}^{r|A_{k}}$, $\mu_{k}^{r|A_{k}}=L_{k}^{r}\mu_{k|k-1}^{r|A_{k}}/c_{2}$.
F5	Fine set fusion	${\hat{x}}_{k}^{A_{k}}=\sum_{r}\mu_{k}^{r|A_{k}}{\hat{x}}_{k}^{r}$, ${\hat{P}}_{k}^{A_{k}}=\sum_{r}\mu_{k}^{r|A_{k}}\left({\hat{P}}_{k}^{r}+\left({\hat{x}}_{k}^{r}-{\hat{x}}_{k}^{A_{k}}\right){\left({\hat{x}}_{k}^{r}-{\hat{x}}_{k}^{A_{k}}\right)}^{⊤}\right)$.
G1	Global probabilities	$\mu_{k|k-1}^{r}=\mu_{k|k-1}^{r|A_{k}}\mu_{k-1}^{A_{k}}$ (Equation 66); $\mu_{k}^{A_{k}}=\left(c_{2}\mu_{k-1}^{A_{k-1}}\right)/\left(c_{1}\mu_{k-1}^{M_{k-1}^{c}}+c_{2}\mu_{k-1}^{A_{k}}\right)$; $\mu_{k}^{M_{k}^{c}}=1-\mu_{k}^{A_{k}}$.
G2	Global fusion (state)	${\hat{x}}_{k}=\mu_{k}^{M_{k}^{c}}{\hat{x}}_{k}^{M_{k}^{c}}+\mu_{k}^{A_{k}}{\hat{x}}_{k}^{A_{k}}$, ${\hat{P}}_{k}=\mu_{k}^{M_{k}^{c}}{\hat{P}}_{k}^{M_{k}^{c}}+\mu_{k}^{A_{k}}{\hat{P}}_{k}^{A_{k}}$.

**Table 3 sensors-25-06621-t003:** ARMSE comparison under Gaussian noise in different scenarios.

Scenario	Method	Position (m)	Velocity (m/s)	Acceleration (m/s^2^)
Quasi-equilibrium Glide	HAMM-MEE	137.2320	48.1836	9.8616
	IMM	161.5792	80.8644	24.0227
	BMA	155.6746	71.5361	19.4388
	HAMM-CKF	136.9951	48.1121	9.8641
Jump Glide	HAMM-MEE	113.1800	41.5979	8.6980
	IMM	132.4838	69.0319	21.3183
	BMA	127.3080	60.7622	17.1564
	HAMM-CKF	112.8430	41.5102	8.7074
Penetration Maneuver	HAMM-MEE	157.5364	53.2733	10.8696
	IMM	185.7967	89.1551	25.5169
	BMA	179.6320	79.3338	20.8543
	HAMM-CKF	156.4991	52.7344	10.7864

**Table 4 sensors-25-06621-t004:** Comparison of average execution time per filtering cycle.

Method	Proposed Method CKF	IMM	BMA	HAMM-CKF
Time (s)	0.0203	0.1025	0.0258	0.0102

**Table 5 sensors-25-06621-t005:** ARMSE comparison under heavy-tailed noise in different scenarios.

Scenario	Method	Position (m)	Velocity (m/s)	Acceleration (m/s^2^)
Quasi-equilibrium Glide	HAMM-MEE	216.4382	107.2747	31.8554
	IMM	271.6558	145.7894	46.9492
	BMA	246.3115	129.7225	41.0685
	HAMM-CKF	240.3056	131.7618	43.4616
Jump Glide	HAMM-MEE	178.4174	91.8257	28.5800
	IMM	219.5540	121.8200	41.2441
	BMA	199.6151	109.4294	36.4410
	HAMM-CKF	194.4407	110.8057	38.4076
Penetration Maneuver	HAMM-MEE	250.2486	119.1319	34.0385
	IMM	309.5043	159.6728	49.2976
	BMA	282.6133	143.2702	43.5457
	HAMM-CKF	275.1503	144.9546	45.8658

**Table 6 sensors-25-06621-t006:** ARMSE comparison under impulsive noise in different scenarios.

Scenario	Method	Position (m)	Velocity (m/s)	Acceleration (m/s^2^)
Quasi-equilibrium Glide	HAMM-MEE	226.8411	112.3787	33.3385
	IMM	283.0154	151.6903	48.8372
	BMA	254.0933	130.8416	40.6520
	HAMM-CKF	248.6504	134.5893	43.9018
Jump Glide	HAMM-MEE	187.2057	96.5216	30.0357
	IMM	230.1583	127.8589	43.2772
	BMA	206.9937	110.8298	36.1546
	HAMM-CKF	203.3353	114.4099	39.2269
Penetration Maneuver	HAMM-MEE	260.2332	124.0035	35.4011
	IMM	322.6418	166.2129	51.3371
	BMA	292.1901	144.4903	43.0166
	HAMM-CKF	285.5285	148.4680	46.4684

**Table 7 sensors-25-06621-t007:** ARMSE comparison under varying sampling rates.

Scenario	Method	Position (m)	Velocity (m/s)	Acceleration (m/s^2^)
Quasi-equilibrium Glide	HAMM-MEE	166.7787	55.8755	10.7591
	IMM	195.1847	92.1058	25.5544
	BMA	186.9404	80.7245	20.4510
	HAMM-CKF	166.2928	55.8301	10.7425
Jump Glide	HAMM-MEE	156.9764	54.4403	10.1176
	IMM	181.9219	86.8043	23.9322
	BMA	176.2706	77.7745	19.4474
	HAMM-CKF	157.0802	54.7887	10.1711
Penetration Maneuver	HAMM-MEE	211.6369	65.9453	11.6019
	IMM	247.0984	108.5975	27.7237
	BMA	236.2371	94.7931	22.0826
	HAMM-CKF	211.2154	65.8220	11.5344

## Data Availability

Data are contained within the article.

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
