# Peer review of "A Hierarchical Adaptive Moment Matching Multiple Model Tracking Method for Hypersonic Glide Target Under Measurement Uncertainty"

_sensors, 2025, doi:10.3390/s25216621_

Round 1

Reviewer 1 Report

Comments and Suggestions for Authors

This paper proposes HAMM (Hierarchical Adaptive Moment Matching), a hierarchical multiple-model tracking framework that adaptively prunes and refines model sets (via multi-frame posterior evolution, Rényi divergence, and moment matching) and couples them with a minimum-error-entropy cubature Kalman filter (MEECKF) to robustly track hypersonic glide targets under complex maneuvers and non-Gaussian measurement noise, improving positioning accuracy and convergence.

  1. It is recommended to include a figure illustrating the tracking model.
  2. Equations 12 and 13 are not clear, and need clarification.
  3. Is the assumption that acceleration components are independent across spatial dimensions valid for real HGT trajectories, and how much does violating it hurt performance? Are the chosen parametric descriptors (αᵢ, βᵢ, σᵢ) and the three discrete mode classes (N/R/P) identifiable and expressive enough under high noise and limited sensor observability?
  4. Figure 1 needs detailed information.
  5. Are the three designed scenarios and the chosen radar placements sufficiently representative of realistic engagement geometries (and is the coordinate convention for the radar positions clearly defined)? Is the sampling period (T = 0.1 s), max detection range (600 km), and the simplified LOS visualization adequate to capture hypersonic maneuver dynamics and realistic detection phenomena (SNR, occlusions, dropouts)?
  6. Are the RMSE improvements statistically significant and consistent across Monte Carlo trials, or could they be from run-to-run variance? Do the convergence speed and steady-state advantages hold under realistic measurement degradations (different SNRs, missed detections, outliers, and varying sampling rates)? Is the superior performance achieved without disproportionate increases in computational cost or hyperparameter tuning that would make it impractical in real-time systems?
  7. Are the accuracy vs. runtime comparisons fair , were all methods implemented and profiled under the same hardware, code optimizations, and parameter budgets? Do the performance gains stem specifically from the MEE-based adaptive noise handling, or from other factors (model-set construction, hyperparameter tuning, or moment-matching)? How well does HAMM-MEECKF generalize to other non-Gaussian noise types?
  8. The Conclusion section should be strengthened, and additional experimental results should be provided to support the claims.

Reviewer 2 Report

Comments and Suggestions for Authors

The paper tackles radar tracking for hypersonic glide targets using a hierarchical multiple-model framework with moment matching and an MEE-based CKF. The problem is well-motivated, the abstract and introduction are generally clear, and the related work is thorough. The mathematical development is detailed and carefully presented. Overall, the manuscript is technically solid and reads well; a few focused language and presentation edits would further enhance clarity and consistency.
1. The particle-filter discussion is accurate but longer than necessary for the introduction. Please condense it to the essentials and keep the focus on why PF is not the preferred choice here, given the radar setting and the later MEECKF design.
2. Equations (9) and (10) appear structurally identical. Consider presenting them under a single template—similar to Eq. (8)—and then specifying parameter choices or special cases.
3. Please avoid very long acronyms in the simulation section (e.g., “HAMM-MEECKF”). Use the full method name on first mention and then refer to it with a concise descriptor such as “the proposed method.”
4.It is advisable to consult relevant operational or defense-domain experts to further validate the plausibility of the data and the methodology.

Reviewer 3 Report

Comments and Suggestions for Authors

This article proposes a hierarchical adaptive moment matching multiple model tracking method for hypersonic glide target under measurement uncertainty. This article is well organized and written, while I have some suggestions before this article can be considered for publication.

1. In Section 1, the authors point out that their contributions include a systematically constructed kinematic tracking model, a hierarchical adaptive strategy for the model set and a robust filter named MEECKF. The authors should explain the differences or improvements against the existing results from these three aspects.

2. In Section 2, the improvements on the kinematic tracking mode should be stressed. If no improvements are made on the tracking model, this part could be described in a more concise way. And an explanation on the consistency between the dynamic model and the proposed/utilized kinematic model of hypersonic glide vehicle should be analyzed.

3. In Subsection 2.2, a novel model set is constructed. I suggest the authors give some examples of different maneuver modes, i.e. draw some typical trajectories.

4. The pseudocode or detailed procedure of the proposed algorithm should be summarized in a table.

5. The proposed robust filter in Section 4 seems to use the M-estimation theory and similar with the existing methods based on M-estimation theory. The authors should clearly describe the main difference or just borrow the idea and use it within the multi-model framework.

6. I suggest the authors check the whole paper and unify the format of the mathematical symbols.

7. There are some grammar errors. The language can be improved by an English native speaker from the beginning to end.

Round 2

Reviewer 1 Report

Comments and Suggestions for Authors

The authors have incorporated the necessary revisions into the manuscript. However, figures and tables should appear after the relevant text and be placed within the same section. In addition, the presentation of the results could be further improved.

Reviewer 3 Report

Comments and Suggestions for Authors

The authors have addressed all my concerns. I suggest the paper be acceptable.

Author Response

Comment1:The authors have addressed all my concerns. I suggest the paper be acceptable.
Response1:We appreciate your positive comments and recommendation.